# Quantitative proteomics reveals the selectivity of ubiquitin-binding autophagy receptors in the turnover of damaged lysosomes by lysophagy

Vinay V Eapen[1,2†‡], Sharan Swarup[1,2†], Melissa J Hoyer[1,2†], Joao A Paulo[1], J Wade Harper[1,2]*

[1]Department of Cell Biology, Harvard Medical School, Boston, Boston, United States; [2]Aligning Science Across Parkinson's (ASAP) Collaborative Research Network, Chevy Chase, United States

*For correspondence: wade_harper@hms.harvard.edu

†These authors contributed equally to this work

Present address: ‡Casma Therapeutics, Cambridge, United States

**ABSTRACT** Removal of damaged organelles via the process of selective autophagy constitutes a major form of cellular quality control. Damaged organelles are recognized by a dedicated surveillance machinery, leading to the assembly of an autophagosome around the damaged organelle, prior to fusion with the degradative lysosomal compartment. Lysosomes themselves are also prone to damage and are degraded through the process of lysophagy. While early steps involve recognition of ruptured lysosomal membranes by glycan-binding galectins and ubiquitylation of transmembrane lysosomal proteins, many steps in the process, and their interrelationships, remain poorly understood, including the role and identity of cargo receptors required for completion of lysophagy. Here, we employ quantitative organelle capture and proximity biotinylation proteomics of autophagy adaptors, cargo receptors, and galectins in response to acute lysosomal damage, thereby revealing the landscape of lysosome-associated proteome remodeling during lysophagy. Among the proteins dynamically recruited to damaged lysosomes were ubiquitin-binding autophagic cargo receptors. Using newly developed lysophagic flux reporters including Lyso-Keima, we demonstrate that TAX1BP1, together with its associated kinase TBK1, are both necessary and sufficient to promote lysophagic flux in both HeLa cells and induced neurons (iNeurons). While the related receptor Optineurin (OPTN) can drive damage-dependent lysophagy when overexpressed, cells lacking either OPTN or CALCOCO2 still maintain significant lysophagic flux in HeLa cells. Mechanistically, TAX1BP1-driven lysophagy requires its N-terminal SKICH domain, which binds both TBK1 and the autophagy regulatory factor RB1CC1, and requires upstream ubiquitylation events for efficient recruitment and lysophagic flux. These results identify TAX1BP1 as a central component in the lysophagy pathway and provide a proteomic resource for future studies of the lysophagy process.

## Introduction

The lysosome—a membrane-bound compartment containing proteolytic enzymes—has several indispensable functions in eukaryotic cells, including a central role in protein homeostasis (*Perera and Zoncu, 2016*; *Saftig and Puertollano, 2021*). First, lysosomes play key roles in the degradation and recycling of proteins delivered from the endocytic, phagocytic, and secretory/biosynthetic pathways. Second, lysosomes are the terminal receptacle for a form of protein and organelle turnover called autophagy. In this process, double membrane structures called autophagosomes are built around cargo through a multistep process, culminating in the closure of the autophagosome around the cargo. Closed autophagosomes then fuse with lysosomes, thereby delivering their cargo to the

lysosomal lumen for degradation (*Yim and Mizushima, 2020*). Third, the lysosome serves as a platform for sensing intracellular (cytosolic and intralysosomal) amino acid availability through regulation of the MTOR–MLST8–RPTOR complex by the Ragulator complex on the lysosomal membrane, including amino acids from both endocytic and autophagic pathways (*Saxton and Sabatini, 2017*). A central element in lysosomal function is the acidification of the organelle during maturation, which promotes the activation of luminal proteolytic enzymes. Processes that lead to damaged lysosomal membranes—including physiological and pathophysiological pathways—can promote loss of the appropriate pH gradient and defective proteostasis. As such, mechanisms have evolved to both repair specific types of membrane damage or in some circumstances promote degradation of the damaged organelle by a process referred to as lysophagy (*Maejima et al., 2013*; *Papadopoulos and Meyer, 2017*; *Papadopoulos et al., 2020*; *Yim and Mizushima, 2020*).

Several mechanisms for recognition of damaged membrane-bound organelles for selective autophagy have been identified. These processes—referred to as organellophagy—generally fall into two classes—ubiquitin (Ub)-dependent and Ub-independent (*Anding and Baehrecke, 2017*; *Khaminets et al., 2016*). In Ub-independent forms of organellophagy (e.g. ER-phagy), receptor proteins typically embedded in the cognate membrane are directly recognized by the autophagic machinery to facilitate engulfment in response to regulatory signals. These cargo receptors employ LC3-interacting region (LIR) motifs to associate with the LIR-docking site (LDS) in one or more of six ATG8 adaptor proteins that are located in the growing autophagosomal membrane by virtue of attachment to phosphatidylethanolamine via their C-terminal glycine residue (*Gubas and Dikic, 2021*; *Johansen and Lamark, 2020*). In contrast, Ub-dependent forms of organellophagy frequently employ a multistep process involving: (1) sensing of organelle damage, (2) ubiquitylation of one or more proteins associated with the membrane of the damaged organelle, (3) recruitment of one or more Ub-binding autophagy receptors containing LIR or other motifs that recruit autophagic machinery, and (4) expansion of the autophagic membrane around the organelle, thereby facilitating delivery to the lysosome (*Gubas and Dikic, 2021*; *Johansen and Lamark, 2020*; *Khaminets et al., 2016*; *Lamark and Johansen, 2021*). This pathway is perhaps best understood in the context of damaged mitochondria, where the Parkin Ub ligase catalyzes ubiquitylation of mitochondrial outer membrane proteins, followed by recruitment of multiple Ub-binding autophagy receptors including Optineurin (OPTN), CALCOCO2 (also called NDP52), SQSTM1 (also called p62), TAX1BP1, and NBR1 to the outer membrane (*Harper et al., 2018*; *Heo et al., 2015*; *Lazarou et al., 2015*; *Moore and Holzbaur, 2016*; *Ordureau et al., 2014*; *Ordureau et al., 2018*; *Ordureau et al., 2020*; *Pickrell and Youle, 2015*; *Richter et al., 2016*; *Wong and Holzbaur, 2014*). However, available data suggest that only OPTN and to a lesser extent CALCOCO2 are necessary for ultimate delivery of damaged mitochondria to lysosomes in the majority of cell types examined thus far (*Heo et al., 2015*; *Lazarou et al., 2015*; *Evans and Holzbaur, 2020*). The mechanistic basis for the utilization of distinct Ub-binding autophagy receptors for specific types of organellophagy is largely unknown, but a common feature appears to be a role for TBK1-dependent phosphorylation of the receptor and/or other components at the 'synapse' between the autophagosome and target organelle (*Harding et al., 2021*; *Heo et al., 2015*; *Heo et al., 2019*; *Moore and Holzbaur, 2016*; *Richter et al., 2016*; *Wild et al., 2011*). In the case of OPTN, its phosphorylation by TBK1 promotes association with Ub chains in the context of mitophagy (*Heo et al., 2015*; *Richter et al., 2016*).

Previous studies have begun to map out pathways by which damaged lysosomes are selected for lysophagy (*Figure 1A*). In response to rupture of the lysosomal membrane, specific galectins (principally LGALS3 and LGALS8, but LGALS1 and LGALS9 have also been shown to be recruited) bind to glycosylated luminal domains of lysosomal transmembrane proteins, while ubiquitylation of lysosomal membrane proteins occurs with kinetics similar to that of galectin recruitment to initiate lysophagy (*Aits et al., 2015*; *Maejima et al., 2013*; *Jia et al., 2018*). Multiple steps in the ubiquitylation process that have been proposed include: (1) assembly of K63-linked and subsequently K48-linked Ub chains on lysosomal proteins, and (2) removal of K48-linked conjugates by the p97 (also called VCP) AAA$^+$ ATPase in combination with the deubiquitylating enzyme YOD1 (*Papadopoulos et al., 2017*). Depletion of the E2 Ub-conjugating enzyme UBE2QL1 dramatically reduces the extent of K48 Ub chain synthesis and impairs K63 Ub chain synthesis (*Koerver et al., 2019*). Multiple E3 Ub ligases have been proposed to ubiquitylate the lysosomal surface, including SCF$^{FBXO27}$ and the LGALS3-binding TRIM16 RING E3, but precisely how these E3s promote the pathway is poorly understood (*Chauhan*

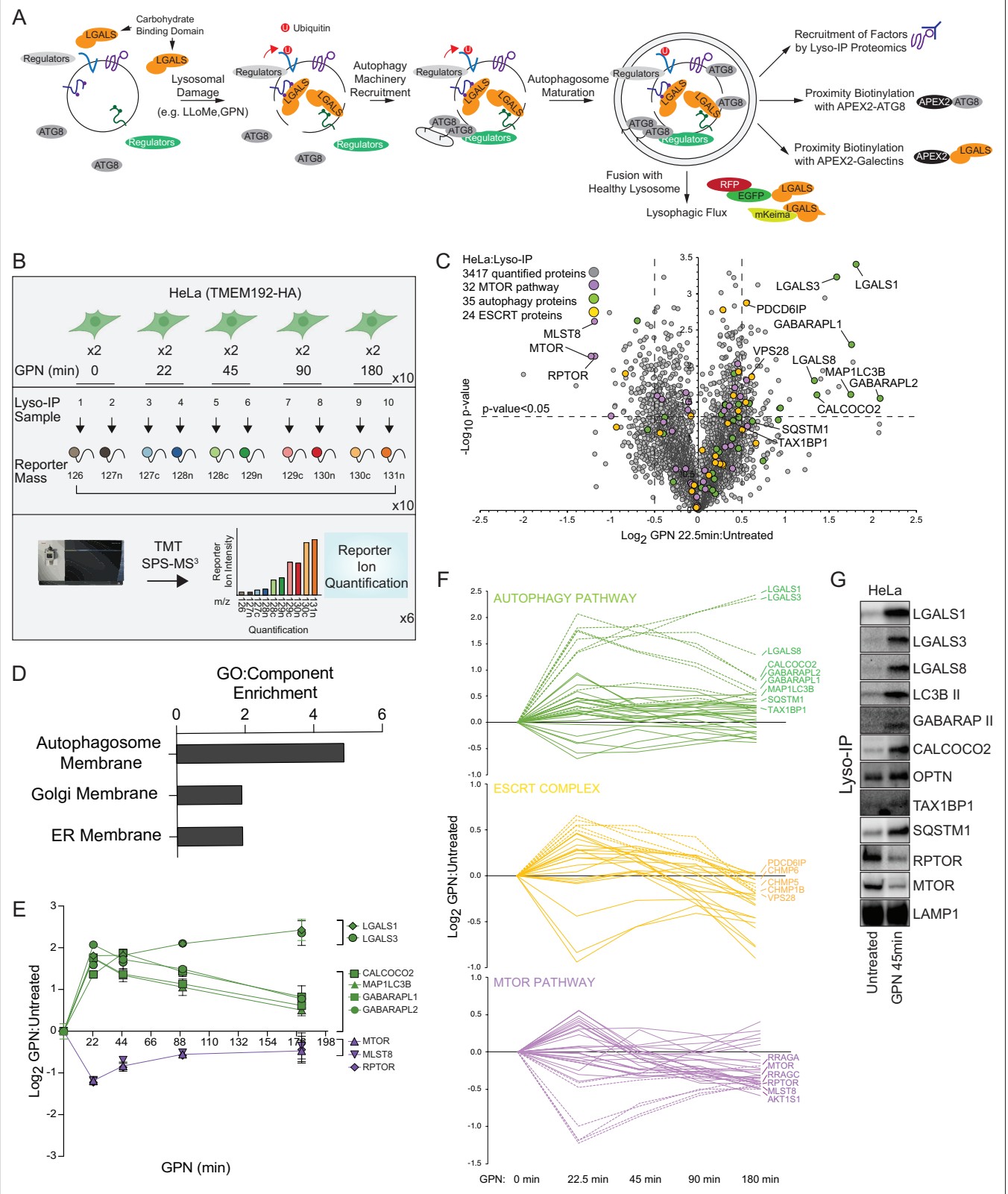

**Figure 1.** Quantitative analysis of the lysosomal proteome in response to damage. (**A**) Scheme depicting major steps in lysophagy and the approaches employed to elucidate components of the pathway. (**B**) Scheme for tandem mass tagging (TMT)-based proteomics of lysosomes from HeLa cells in response to lysosome rupture by glycyl-ʟ-phenylalanine 2-naphthylamide (GPN). Cells expressing TMEM192-HA were left untreated or treated with GPN for the indicated period of time (in duplicate) and cell lysates subjected to a Lyso-IP protocol prior to TMT-based proteomics. (**C**) Volcano plot for GPN

*Figure 1 continued on next page*

*Figure 1 continued*

(22.5 min)-treated cells versus untreated Lyso-IP samples (Log$_2$ FC versus −Log$_{10}$ p-value) based on the TMT experiment in (B). Specific categories of proteins are indicated by colored circles. (**D**) GO enrichment (component) for proteins that accumulate on lysosomes in response to GPN treatment. (**E**) Time course reflecting the dynamics of recruitment or loss of selected proteins from lysosomes in response to GPN treatment. Error bars represent SD from two biological replicates. (**F**) Dynamics of recruitment or loss of proteins linked with autophagy (top), ESCRT (middle), and MTOR (lower) pathways in association with lysosomes upon GPN treatment. All the lines for each category represent individual proteins (see *Supplementary file 2*), and proteins with the most highly dynamic changes are indicated as dashed lines. (**G**) HeLa cells were either left untreated or treated with GPN for 45 min prior to isolation of lysosomes by Lyso-IP. Samples were then subjected to immunoblotting with the indicated antibodies.

The online version of this article includes the following figure supplement(s) for figure 1:

**Source data 1.** Uncropped blots for *Figure 1G*.

**Source data 2.** TMT ratios from GPN time course for *Figure 1E*.

**Figure supplement 1.** Quantitative analysis of the lysosomal proteome in response to damage.

**Figure supplement 1—source data 1.** Uncropped blots for *Figure 1B, D and H*.

*et al., 2016*; *Yoshida et al., 2017*). Some aspects of this pathway have parallels with xenophagy, the process by which intracellular bacteria is removed by autophagy, including early ubiquitylation steps and recruitment of LGALS8 (*Thurston et al., 2009*; *Thurston et al., 2012*). In addition, multiple Ub-binding cargo receptors including OPTN, CALCOCO2, and TAX1BP1 are recruited to ubiquityl-ated bacteria and are required for efficient xenophagy in various contexts (*Thurston et al., 2009*; *Thurston et al., 2012*; *Tumbarello et al., 2015*; *Wild et al., 2011*). While multiple cargo receptors have been reported to be recruited to damaged lysosomes (*Bussi et al., 2018*; *Davis et al., 2021*; *Koerver et al., 2019*), the underlying mechanisms for recruitment, the identity of the receptors critical for lysophagic flux, and the reasons for the diversity of receptors that are recruited remain unknown.

In this study, we set out to systematically examine lysophagy using a series of complementary proteomic approaches with the goal of identifying previously unrecognized machinery required for lysophagic flux. Using lysosomal immunoprecipitation (Lyso-IP) in the context of lysosomal membrane damage, we identified several Ub-binding cargo receptors and ATG8 proteins that are rapidly recruited to these organelles, and verified that LGALS1, LGALS3, and LGALS8 are recruited as well. Parallel APEX2-driven proximity biotinylation experiments using ATG8 proteins and specific galectins identi-fied a cohort of lysosomal proteins, ESCRT III complex members, and autophagy regulatory proteins that are dramatically enriched during lysophagic flux, including specific Ub-binding cargo receptors. In order to assess the functional roles of various components in lysophagy, we developed LGALS3-based fluorescent flux reporters that monitor delivery of damaged lysosomes to healthy lysosomes via auto-phagy. We systematically examined the requirement of cargo receptors that are recruited to damaged lysosomes in HeLa cells, and found that while cells lacking TAX1BP1 were completely deficient for lysophagy, cells lacking OPTN, CALCOCO2, or SQSTM1 still maintained significant lysophagic flux, indicating that TAX1BP1 plays a critical nonredundant function. Similarly, human embryonic stem (ES) cell-derived induced neurons (iNeurons) lacking TAX1BP1 are also defective for lysophagic flux. Lysophagic flux via TAX1BP1 required its N-terminal SKICH domain, as well as Ala-114 within the SKICH domain, which is known to function in the recruitment of both the TBK1 protein kinase via the NAP1 adaptor and the RB1CC1 subunit of the ULK1 kinase complex (*Fu et al., 2018*; *Ohnstad et al., 2020*). Consistent with a role for TBK1, cells lacking TBK1 or addition of a small molecular inhibitor of TBK1 blocks lysophagic flux. Additional experiments indicate that recruitment of TAX1BP1 and OPTN to damaged lysosomes is promoted by an upstream Ub signal. These data provide a resource for factors involved in lysophagy and reveal a unique role for TAX1BP1 in the removal of damaged lyso-somes that appear to be distinct from the mechanisms used for removal of mitochondria downstream of Parkin, which relies primarily on OPTN and CALCOCO2.

## Results

### Quantitative lysosomal proteomics during lysophagy

Previous studies have revealed that lysosomal membrane damage can result in increased ubiqui-tylation of lysosomal proteins as well as the recruitment of specific galectins (*Aits et al., 2015*; *Jia et al., 2018*; *Jia et al., 2020a*; *Jia et al., 2020b*; *Maejima et al., 2013*; *Yoshida et al., 2017*). We

set out to employ a suite of unbiased quantitative proteomics approaches to systematically identify proteins that are dynamically recruited to damaged lysosomes using the well-characterized damaging agents L-Leucyl-L-Leucine methyl ester (LLOMe) or glycyl-L-phenylalanine 2-naphthylamide (GPN) (*Figure 1A*). LLOMe enters the lysosomal system via endocytosis and forms conjugates that can specifically rupture lysosomal membranes on a subset of lysosomes to initiate lysophagy, while GPN promotes lysosomal osmotic swelling and rupture (*Bright et al., 2016*; *Jadot et al., 1984*; *Maejima et al., 2013*; *Skowyra et al., 2018*). To facilitate quantitative identification of candidates, we merged the Lyso-IP approach (*Abu-Remaileh et al., 2017*) with Tandem Mass Tagging (TMT)-based proteomics via synchronous precursor selection (SPS) and quantification of reporter ions using MS$^3$ (*McAlister et al., 2014*). Lysosomes in HeLa cells were tagged by integrating a 3X-HA (HA) tag into the cytosolic C-terminus of TMEM192 gene via CRISPR-Cas9 (*Figure 1—figure supplement 1A-D*) and α-HA immunoprecipitates from these cells yielded an enrichment of transmembrane, luminal, and membrane-associated lysosomal proteins when compared with untagged cells, as shown for the HeLa cell system (*Figure 1—figure supplement 1E-G*, *Supplementary file 1*).

To identify proteins recruited to lysosomes during lysophagy, HeLa$^{TMEM192-HA}$ cells in biological duplicates were left untreated or treated with GPN for 22.5, 45, 90, or 180 min, followed by Lyso-IP and analysis by TMT-MS$^3$ (*Figure 1B* and *Supplementary file 2*). This resulted in the identification of several proteins that were enriched on ruptured lysosomes at one or more time points post-GPN, including multiple ATG8 proteins (MAP1LC3B, GABARAPL1, and GABARAPL2), galectins (LGALS1, LGALS3, and LGALS8), and the Ub-binding cargo receptors (CALCOCO2 and SQSTM1) with Log$_2$ fold change (FC) >0.5 (p-value <0.05) for at least one time point (*Figure 1C–E*, *Figure 1—figure supplement 1E*). TAX1BP1 was also found to be slightly enriched at 22.5 min, but was also found constitutively in undamaged lysosomes (*Figure 1—figure supplement 1E*, *Supplementary files 1 and 2*). Previous studies have indicated that damaged lysosomal membranes may also be subject to repair by components of the ESCRT system (*Jia et al., 2020b*; *Radulovic et al., 2018*; *Skowyra et al., 2018*). Consistent with this, we observed transient enrichment of ESCRT-III components CHMP1B, CHMP5, CHMP6, and PCDC6IP by Lyso-IP (*Figure 1F*). We also observed a reduction in the abundance of the MTORC1 complex (MLST8, RPTOR, and MTOR) post-damage, consistent with previous reports that lysosomal damage leads to loss of this kinase complex from the lysosomal surface (*Figure 1C, E and F*; *Jia et al., 2018*). We verified enrichment of galectins, lipidated forms of MAP1LC3B and GABARAP, OPTN, CALCOCO2, TAX1BP1, and SQSTM1, as well as loss of RPTOR and MTOR, using immunoblotting of Lyso-IP fractions upon lysosomal damage (*Figure 1G*). The recruitment of these candidate Ub-binding cargo receptors is consistent with a previously reported role for lysosomal ubiquitylation in response to rupture (*Koerver et al., 2019*; *Yoshida et al., 2017*).

## Proximity biotinylation of autophagy receptors and galectins during lysosomal damage

The rapid recruitment of ATG8 and galectin proteins to damaged lysosomes (*Figure 1E*) led us to employ APEX2-driven proximity biotinylation as a complementary approach to identify proteins that may link the autophagic machinery with ruptured lysosomes (*Figure 2A and B*). To initially check for fusion protein function, cells stably expressing FLAG-APEX2 fusions with GABARAPL2 (WT or LDS mutant Y49A/L50A) or MAP1LC3B (WT or LDS mutant K51A) (*Mizushima, 2020*; *Figure 2—figure supplement 1A*) were treated for 1 hr with LLOMe prior to immunostaining to examine recruitment of the FLAG-tagged APEX2 protein to lysosomes marked with α-LAMP1 antibodies (*Figure 2—figure supplement 1B*). Both WT constructs formed more numerous and larger puncta than the LDS mutants, consistent with enhanced lysosomal recruitment. We then treated these cells together in biological triplicates (60 min) or duplicate (0 min) with LLOMe (1 hr) in the presence of biotin phenol, followed by H$_2$O$_2$ (1 min), and immediately processed for biotin enrichment and proteomics in two 10-plex TMT experiments (*Figure 2B*, *Supplementary file 3*). From ~1300 proteins identified with APEX2-MAP1LC3B, we identified 46 proteins that were enriched (Log$_2$ FC >1.0; p-value <0.05) in the presence of LLOMe (*Figure 2C*) with only the lysosomal compartment being significantly enriched when compared with several subcellular compartments (*Figure 2D*). Similarly, APEX2-GABARAPL2 was also enriched in autophagy receptors and lysosomal proteins (*Figure 2E and F*), with numerous proteins being in common with APEX2-MAP1LC3B-enriched proteins (*Figure 2G*, *Figure 2—figure supplement 1C*). Five major functional classes of proteins were identified with one or both ATG8 proteins: (1)

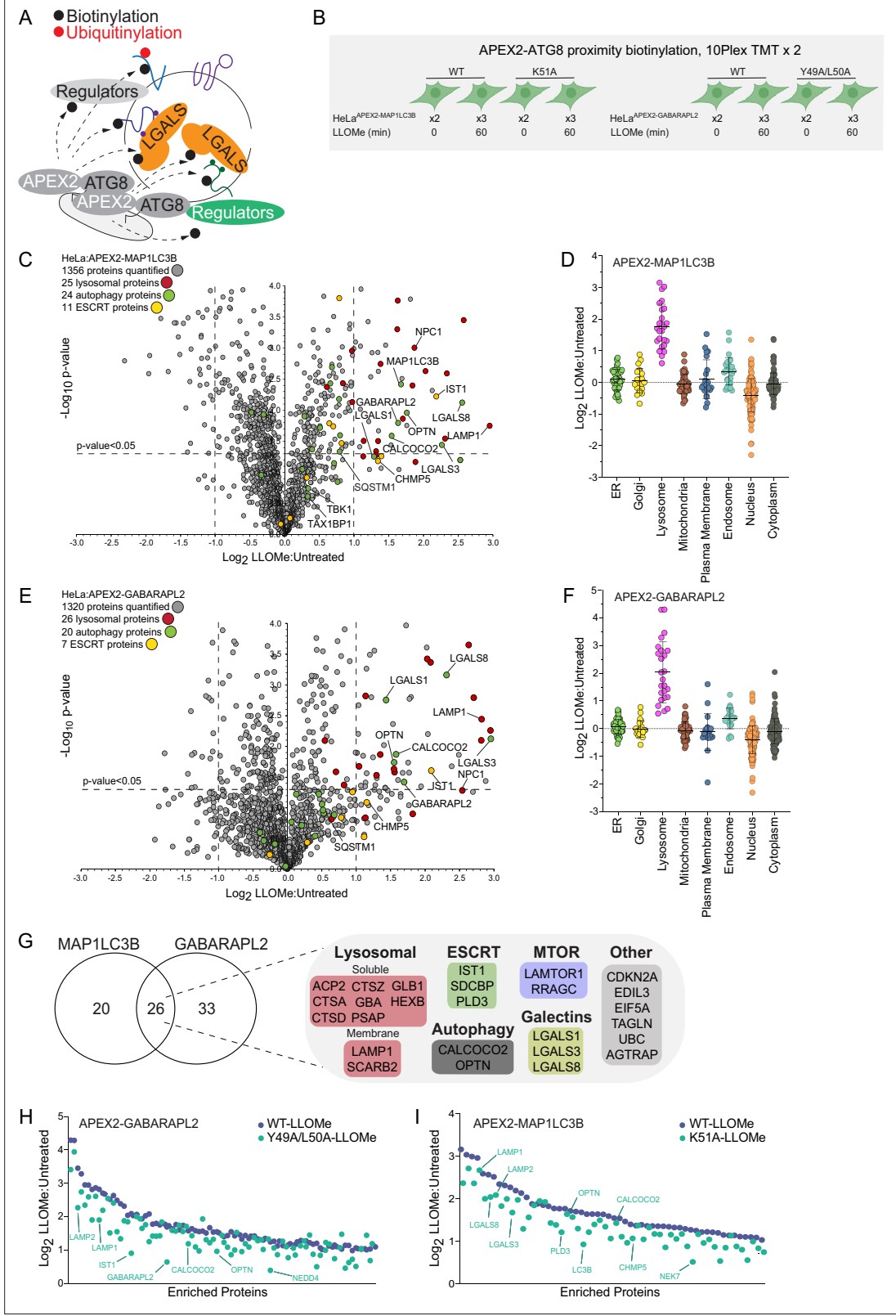

**Figure 2.** Proximity biotinylation of ATG8 proteins in response to lysosomal damage. (**A**) Scheme depicting proximity biotinylation of proteins in response to recruitment of ATG8 proteins to damaged lysosomes. (**B**) Experimental workflow for ATG8 proximity biotinylation. APEX2-tagged GABARAPL2 (or the corresponding Y49A mutant) or MAP1LC3B (or the corresponding K51A mutant) expressed in HeLa cells were subjected to proximity biotinylation 60 min post-LLOMe treatment using 10plex TMT. (**C**) Volcano plot for LLOMe (60 min)-treated cells versus untreated cells ($Log_2$

*Figure 2 continued on next page*

*Figure 2 continued*

FC versus −Log$_{10}$ p-value) for APEX-MAP1LC3B-based proximity biotinylation based on the TMT experiment in B. Specific categories of proteins are indicated by colored circles. (**D**) Log$_2$ FC for individual proteins localized to the indicated subcellular compartments found to be enriched in biotinylated proteins from cells expressing APEX2-MAP1LC3B. Mean and standard deviation are calculated from two untreated and three treated biological replicates. (**E**) Volcano plot for LLOMe (60 min)-treated cells versus untreated cells (Log$_2$ FC versus −Log$_{10}$ p-value) for APEX-GABARAPL2-based proximity biotinylation based on the TMT experiment in (**B**). Specific categories of proteins are indicated by colored circles. (**F**) Log$_2$ FC for individual proteins localized to the indicated subcellular compartments found to be enriched in biotinylated proteins from cells expressing APEX2-GABARAPL2. Mean and standard deviation are calculated from two untreated and three treated biological replicates. (**G**) Summary of overlap between biotinylated proteins found with MAP1LC3B and GABARAPL2 APEX2 proteomics. Proteins enriched with Log$_2$ FC >1.0 and p-value <0.05 were included. Proteins identified in both APEX2 experiments are indicated. (**H**) Plot of means of Log$_2$ FC for biotinylated proteins in cells expressing APEX-GABARAPL2 or the Y49A/L50A mutant. Means are calculated from two untreated and three treated biological replicates. (**I**) Plot of means of Log$_2$ FC for biotinylated proteins in cells expressing APEX-MAP1LC3B or the Y49A/L50A mutant. Means are calculated from two untreated and three treated biological replicates.

The online version of this article includes the following figure supplement(s) for figure 2:

**Source data 1.** Log$_2$ FCs of various organelle proteins for APEX2-MAP1LC3B in response to LLOMe for ***Figure 2D***.

**Source data 2.** Log$_2$ FCs of various organelle proteins for APEX2-GABARAPL2 in response to LLOMe for ***Figure 2F***.

**Source data 3.** Log$_2$ FCs of GABARAPL2 LIR dependent interactors for ***Figure 2H***.

**Source data 4.** Log$_2$ FCs of MAP1LC3B LIR-dependent interactors for ***Figure 2I***.

**Figure supplement 1.** Proximity biotinylation of ATG8 proteins in response to lysosomal damage.

**Figure supplement 1—source data 1.** Uncropped blots for ***Figure 2—figure supplement 1A***.

**Figure supplement 1—source data 2.** Source data for APEX2 LC3 LIR-dependent interactors.

**Figure supplement 1—source data 3.** Source data for APEX2 GABARAPL2 LIR-dependent interactors.

Ub-binding autophagy receptors (OPTN and CALCOCO2), (2) resident lysosomal membrane proteins (LAMP1, LAMP2, and SCARB2), (3) galectins (LGALS1, LGALS3, and LGALS8), (4) luminal resident lysosomal proteins (GBA, HEXB, GLB1, PSAP, PLD3, CTSZ, CTSA, CTSD, CTSC, and GNS), and (5) components of the Ragulator/Lamtor complex (RRAGC and LAMTOR1) known to associate with the cytosolic face of the lysosomal membrane (***Saxton and Sabatini, 2017***). Interestingly, proximity biotinylation of a subset of enriched proteins with GABARAPL2, including OPTN, LAMP1, and LAMP2, was partially dependent upon the presence of an intact LDS, although the effect was much less dramatic with the MAP1LC3B$^{K51A}$ mutant (***Figure 2H, I***, ***Figure 2—figure supplement 1D***,E).

In an orthogonal set of two 11-plex TMT experiments, we performed proximity biotinylation with APEX2-tagged LGALS1, LGALS3, and LGALS8 (***Figure 3A and B*** and ***Supplementary file 4***). Stably expressed APEX2-tagged galectins (***Figure 3—figure supplement 1A***) were recruited to lysosomes in response to LLOMe, based on immunofluorescence in fixed cells, indicating that the APEX2 fusions were functional (***Figure 3—figure supplement 1B***). Similar to the APEX2-ATG8 fusions, APEX2-LGALS1, 3, and 8 all displayed enriched biotinylation of the lysosomal compartment, consistent with the known translocation of galectins to sites of lysosome membrane damage (***Figure 3C–F***). Beyond shared lysosomal enrichment, the proximity interactomes of galectins 1, 3, and 8 displayed key differences. Notably, Gene Ontology (GO) analysis of the galectin interactomes indicated that only LGALS8 showed a clear increased interaction with terms associated with autophagy and MTOR signaling driven by preferential enrichment of RRAGC, LAMTOR1, and LAMTOR2 (***Figure 3G***, ***Figure 4—figure supplement 1A***). The specificity of LGALS8 interactions with members of the MTOR complex is consistent with recent reports demonstrating its role in modulating MTOR signaling during lysosomal stress (***Jia et al., 2018***).

## Lysophagy proteome landscape

In order to develop a lysophagy proteome landscape, we organized proteins that were detected as being enriched by Lyso-IP and proximity biotinylation of ATG8 and galectin proteins based on functional categories (***Figure 4A***, ***Figure 4—figure supplement 1A***). All proximity biotinylation experiments showed strong enrichment for GO terms linked with lysosomes, autophagy, and membrane fusion, among other terms (***Figure 4—figure supplement 1A***). All three galectins were found to associate with a cohort of luminal hydrolytic enzymes (e.g., CTSB and CTSD) and lipid modifying proteins (e.g., GLB1, HEXB, and GBA), indicating that they all likely access the lysosomal lumen upon

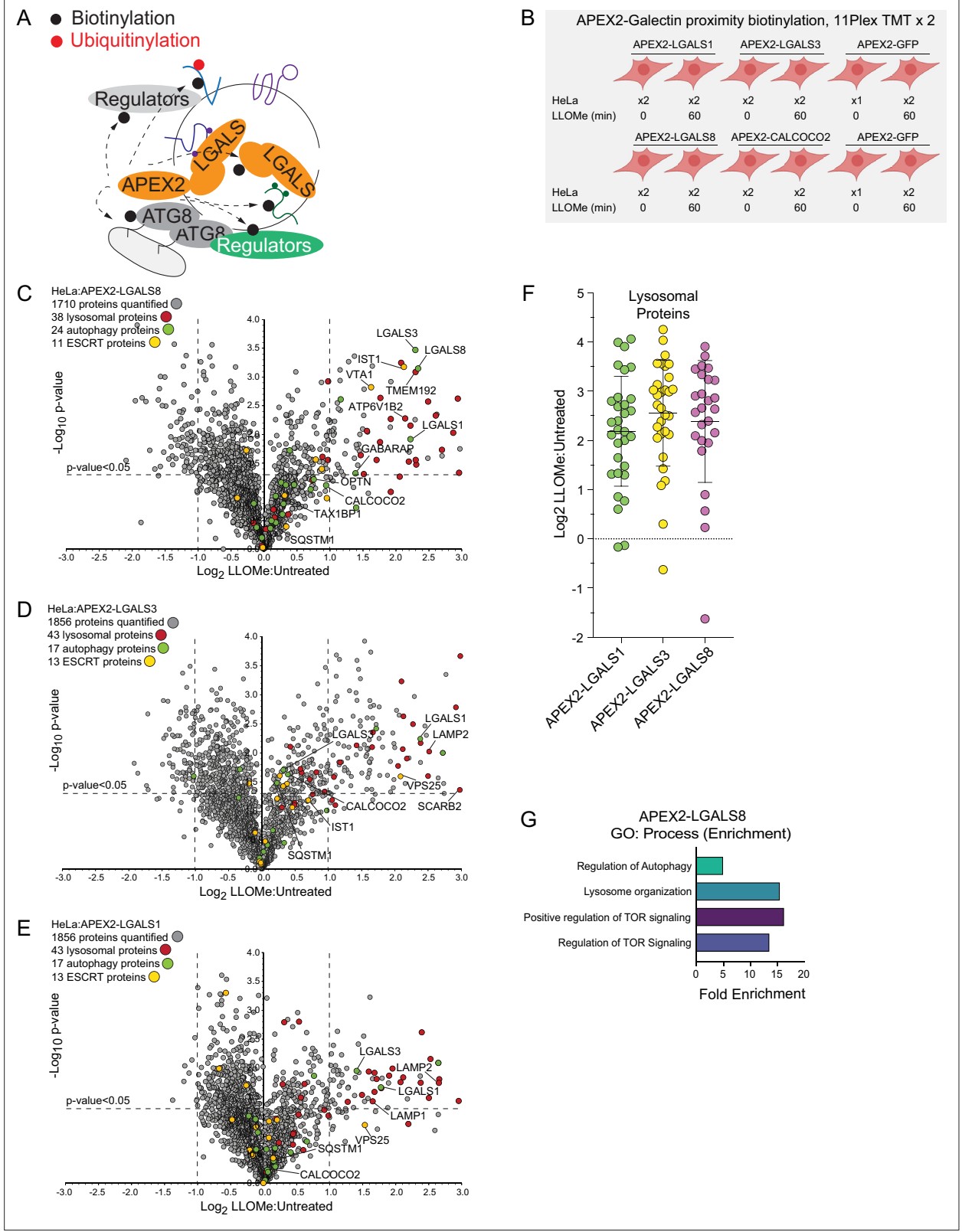

**Figure 3.** Proximity biotinylation of galectins in response to lysosomal damage. (**A**) Scheme depicting proximity biotinylation of proteins in response to recruitment of LGALS1, LGALS3, and LGALS8 proteins to damaged lysosomes. (**B**) Experimental workflow for galectin proximity biotinylation. APEX2-tagged LGALS1, LGALS3, and LGALS8 expressed in HeLa cells were subjected to proximity biotinylation 60 min post-LLOMe treatment using 10-plex TMT. (**C**) Volcano plot for LLOMe (60 min)-treated cells versus untreated cells ($Log_2$ FC versus $-Log_{10}$ p-value) for APEX-LGALS8-based proximity

*Figure 3 continued on next page*

*Figure 3 continued*

biotinylation based on the TMT experiment in (B). Specific categories of proteins are indicated by colored circles. (**D**) Volcano plot for LLOMe (60 min)-treated cells versus untreated cells (Log$_2$ FC versus −Log$_{10}$ p-value) for APEX-LGALS3-based proximity biotinylation based on the TMT experiment in B. Specific categories of proteins are indicated by colored circles. (**E**) Volcano plot for LLOMe (60 min)-treated cells versus untreated cells (Log$_2$ FC versus −Log$_{10}$ p-value) for APEX-LGALS1-based proximity biotinylation based on the TMT experiment in B. Specific categories of proteins are indicated by colored circles. (**F**) Log$_2$ FC for individual proteins localized to the lysosomal compartment found to be enriched in biotinylated proteins from cells expressing the indicated APEX2-galectin protein. Mean and standard deviation are calculated from two untreated and two treated biological replicates. (**G**) GO: process enrichment categories for APEX2-LGALS8.

The online version of this article includes the following figure supplement(s) for figure 3:

**Source data 1.** Log$_2$ FCs for lysosomal proteins from APEX2-LGALS1, 3, and 8 for *Figure 3F*.

**Source data 2.** GO enrichments for APEX2-LGALS8 for *Figure 3G*.

**Figure supplement 1.** Proximity biotinylation of galectins in response to lysosomal damage.

**Figure supplement 1—source data 1.** Uncropped blots for *Figure 3—figure supplement 1A*.

damage (*Figure 4A*). Similarly, both LGALS1 and LGALS8 APEX2 experiments resulted in enrichment of LGALS3, suggesting that individual galectins themselves are in close proximity to each other within damaged lysosomes (*Figure 4A*). Interestingly, all three galectins were found to biotinylate lysosomal membrane proteins LAMP1 and LAMP2, but APEX2-LGALS8 was selectively enriched in CD63, TMEM192, and several V-ATPase subunits (*Figure 4A*). Many proteins found with APEX2-ATG8 proteins were also identified with galectin proximity biotinylation, including both luminal proteins and lysosomal membrane proteins, as well as mTOR regulatory components (*Figure 4A*). Interestingly, although overexpressed LGALS9 has been reported to be recruited to damaged lysosomes and to be required for lysosomal ubiquitylation (*Jia et al., 2018*), we failed to detect endogenous LGALS9 in any of the proteomics experiments performed here. This dataset provides a rich resource for future studies in the lysophagy pathway.

## Multiple Ub-binding autophagy receptors are recruited to damaged lysosomes

Among the proteins found to be enriched by either APEX2-ATG8, APEX2-galectin, or Lyso-IP was the autophagy cargo receptor CALCOCO2, and in some experiments OPTN and TAX1BP1 were also enriched (*Figures 1C, G and 2G*). As such, we systematically examined recruitment of cargo receptor proteins to damaged lysosomes using immunofluorescence (*Figure 4B–E*). As expected, LGALS3 was recruited to LAMP1-positive lysosomes, a subset of which were also positive for MAP1LC3B (*Figure 4B*). In untreated cells, OPTN, TAX1BP1, and CALCOCO2 displayed diffuse localization with little evidence of colocalization with LAMP1-positive lysosomes (Mander's overlap coefficient [MOC] <0.02) (*Figure 4C–H*). However, after addition of LLOMe for 1 hr and after 4 hr washout of LLOMe treatment for 1 hr, there was increased colocalization of these receptors with lysosomes, with TAX1BP1 displaying the most dramatic increase in MOC (a mean of 0.25–0.45 for TAX1BP1) (*Figure 4C–H*). As an independent approach to examine recruitment of cargo receptors to lysosomes, we employed proximity biotinylation of CALCOCO2, TAX1BP1, SQSTM1, and OPTN, and each APEX2-fusion protein was shown to associate with a subset of lysosomes upon damage (*Figure 4—figure supplement 1B-D*). We observed enrichment of numerous specific lysosomal proteins, ESCRT and galectins with CALCOCO2, TAX1BP1, SQSTM1, and/or OPTN-APEX2 proteins 60 min after LLOMe treatment, (*Figure 4—figure supplement 1B-D*; *Supplementary file 5*).

## Measurement of lysophagic flux with Lyso-Keima

While multiple cargo receptors are recruited to damaged lysosomes (*Figure 4C–E*; *Davis et al., 2021*; *Koerver et al., 2019*; *Papadopoulos et al., 2017*), to date, the cargo receptors critical for linking damaged lysosomes to the core autophagy machinery have not been clearly delineated, although knockdown of SQSTM1 by RNAi has been reported to result in reduced lysophagic flux (*Papadopoulos et al., 2017*). We therefore systematically probed cargo receptor involvement in lysophagy. We first developed a tool for the quantitative assessment of lysophagic flux by employing monomeric mKeima (referred to here as Keima) fused with LGALS3, which we term Lyso-Keima (*Figure 5A*). Keima is a pH-responsive reporter that undergoes a chromophore resting charge-state change upon

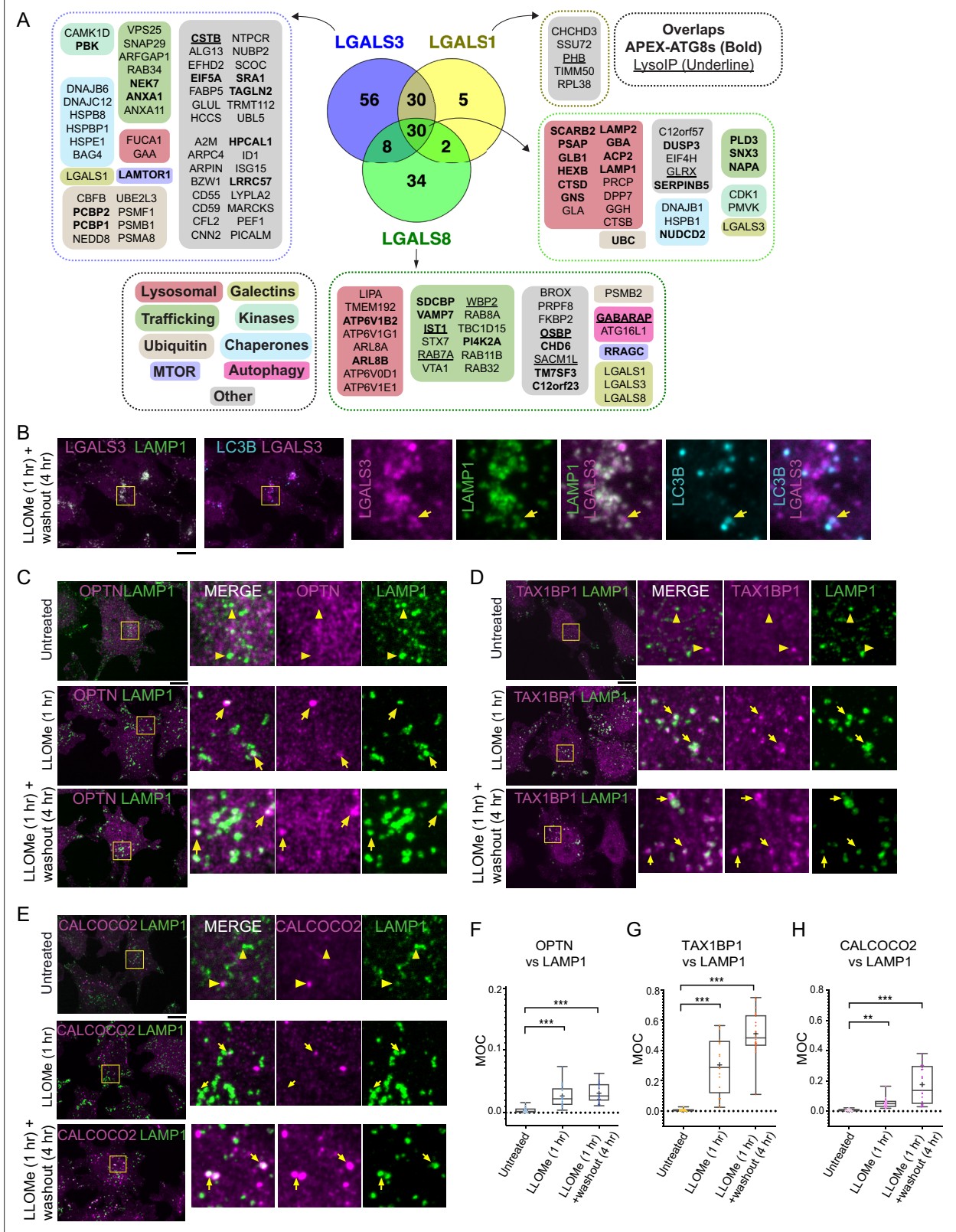

**Figure 4.** Landscape of lysophagy reveals autophagy receptor recruitment. (**A**) Summary of proteins in proximity to galectins and integration with associations found with APEX2-ATG8 (bold) and Lyso-IP (underline). Other functional classes are indicated. (**B**) Localization of LGALS3 with LAMP1 and MAP1LC3B in response to lysosomal damage. Cells were treated with LLOMe for 1 hr and the LLOMe washed out for 4 hr prior to immunofluorescence using the indicated antibodies and imaging by confocal microscopy. Scale bars 10 μm. Zoom-in panels, 10 μm × 10 μm. (**C**) Cells were left untreated

*Figure 4 continued on next page*

*Figure 4 continued*

(Untreated), treated with LLOMe for 1 hr and either fixed (LLOMe 1 hr) or the LLOMe was washed out for 4 hr prior to fixation (LLOMe 1 hr+ washout 4 hr). Immunofluorescence was done using α-OPTN/α-LAMP1 and imaging by confocal microscopy. Scale bars 10 μm. Zoom-in panels, 10 μm × 10 μm. (**D**) Cells were treated as in (C). Immunofluorescence was done using α-TAX1BP1/α-LAMP1 and imaging by confocal microscopy. Scale bars 10 μm. Zoom-in panels, 10 μm × 10 μm. (**E**) Cells were treated as in (C). Immunofluorescence was done using α-CALCOCO2/α-LAMP1 and imaging by confocal microscopy. Scale bars 10 μm. Zoom-in panels, 10 μm × 10 μm. (**F**) Quantification of OPTN localization at LAMP1 lysosomes using Mander's overlap coefficient (MOC). 23 (0 hr), 19 (1 hr), and 22 (4 hr washout) cells were analyzed for MOC. ***p < 0.001. + marks the mean and the line marks the median. The plot represents merged data from three biological replicates for each condition. (**G**) Quantification of TAX1BP1 localization at LAMP1 lysosomes using Mander's overlap coefficient (MOC). 20 (0 hr), 17 (1 hr), and 22 (4 hr washout) cells were analyzed for MOC. ***p < 0.001. + marks the mean and the line marks the median. The plot represents merged data from three biological replicates for each condition. (**H**) Quantification of CALCOCO2 localization at LAMP1 lysosomes using MOC. 18 (0 hr), 20 (1 hr), and 21 (4 hr washout) cells were analyzed for MOC. **p < 0.01 and ***p < 0.001. + marks the mean and the line marks the median. The plot represents merged data from three biological replicates for each condition.

The online version of this article includes the following source data and figure supplement(s) for figure 4:

**Source data 1.** Mander's overlap coefficient (MOC) values for *Figure 4F, G and H*.

**Figure supplement 1.** Proximity biotinylation of Ub-binding cargo adaptors in response to lysosomal rupture.

**Figure supplement 1—source data 1.** Uncropped blots for *Figure 4—figure supplement 1B*.

trafficking to the lysosome (pH of ~4.5) and is stable within the lysosome, allowing flux measurements by flow cytometry or microscopy (*Katayama et al., 2011*). The Keima protein itself is also stable to lysosomal proteases, and the appearance of a 'processed' Keima protein by immunoblotting therefore reveals lysosomal trafficking of the Keima fusion protein (*An and Harper, 2018*; *Katayama et al., 2011*). HeLa cells expressing Keima-LGALS3 were treated with LLOMe (1 hr) and then chased with fresh media for the indicated time prior to analysis by imaging, flow cytometry, or immunoblotting for 'processed' Keima (*Figure 5B–G*). Under basal conditions, Keima-LGALS3 was diffusely cytosolic with signal observed only in the 442 nm excitation channel (neutral pH) (*Figure 5B*). However, after the LLOMe chase (1 hr LLOMe treated and 4 –12 hr washout), we observed a dramatic relocalization of the reporter into puncta in the 442 nm channel, consistent with recruitment to damaged lysosomes. Importantly, a large fraction of these puncta displayed a signal ratio greater than one when comparing the 561 nm/442 nm ratio at 12 hr washout, indicative of trafficking of damaged lysosomes into an acidic compartment (*Figure 5B–D*). The presence of acidic Keima-LGALS3 puncta was completely blocked by the addition of Bafilomycin A1 (a lysosomal acidification inhibitor, BafA) during the washout (*Figure 5C–D*). This ratio shift in LLOMe chased Keima-LGALS3 cells could also be measured using flow cytometry analysis; the 561 nm/488 nm excitation ratio was increased approximately twofold 1 hr after LLOMe and washout (12 hr) in biological triplicate assays (*Figure 5E*). This increase was completely blocked by incubation of cells with BafA during the LLOMe washout (*Figure 5E*, *Figure 5—figure supplement 1A*). These results are consistent with the hypothesis that damaged Keima-positive lysosomes are trafficked to healthy lysosomes, potentially including those newly generated via the TFEB-mediated transcriptional response, for elimination (*Nakamura et al., 2020*).

Previous studies have indicated that lysophagy—as monitored by loss of galectin-positive puncta—requires both p97 activity and the Ub system (*Papadopoulos et al., 2017*). To further validate Keima-LGALS3 flux, we examined the effect of inhibition of the ubiquitin E1-activating enzyme (UBA1) using the TAK243 small molecule (E1i) (*Hyer et al., 2018*) and the p97 inhibitor CB-5083 (p97i) (*Anderson et al., 2015*). TAK243 completely blocked Keima-LGALS3 flux as assessed by both flow cytometry and the Keima-processing assay (*Figure 5F* and *Figure 5—figure supplement 1B*), while p97i blocked flux to an extent similar to that see with a small molecule inhibitor of ULK1 (ULK1i) (*Figure 5F*). Finally, we examined the time course of Keima-LGALS3 flux into the lysosome by using the Keima processing assay (*Figure 5G*). Cells were treated with LLOMe for 1 hr, washed and extracts from cells harvested at the indicated times were subjected to immunoblotting with α-Keima antibodies. Processed Keima was detected as early as 4 hr post-washout, reached maximal levels at 6 hr, and was maintained through 12 hr (*Figure 5G*). Taken together, these data indicate that Keima-LGALS3 can be used to monitor lysophagic flux using multiple assay formats.

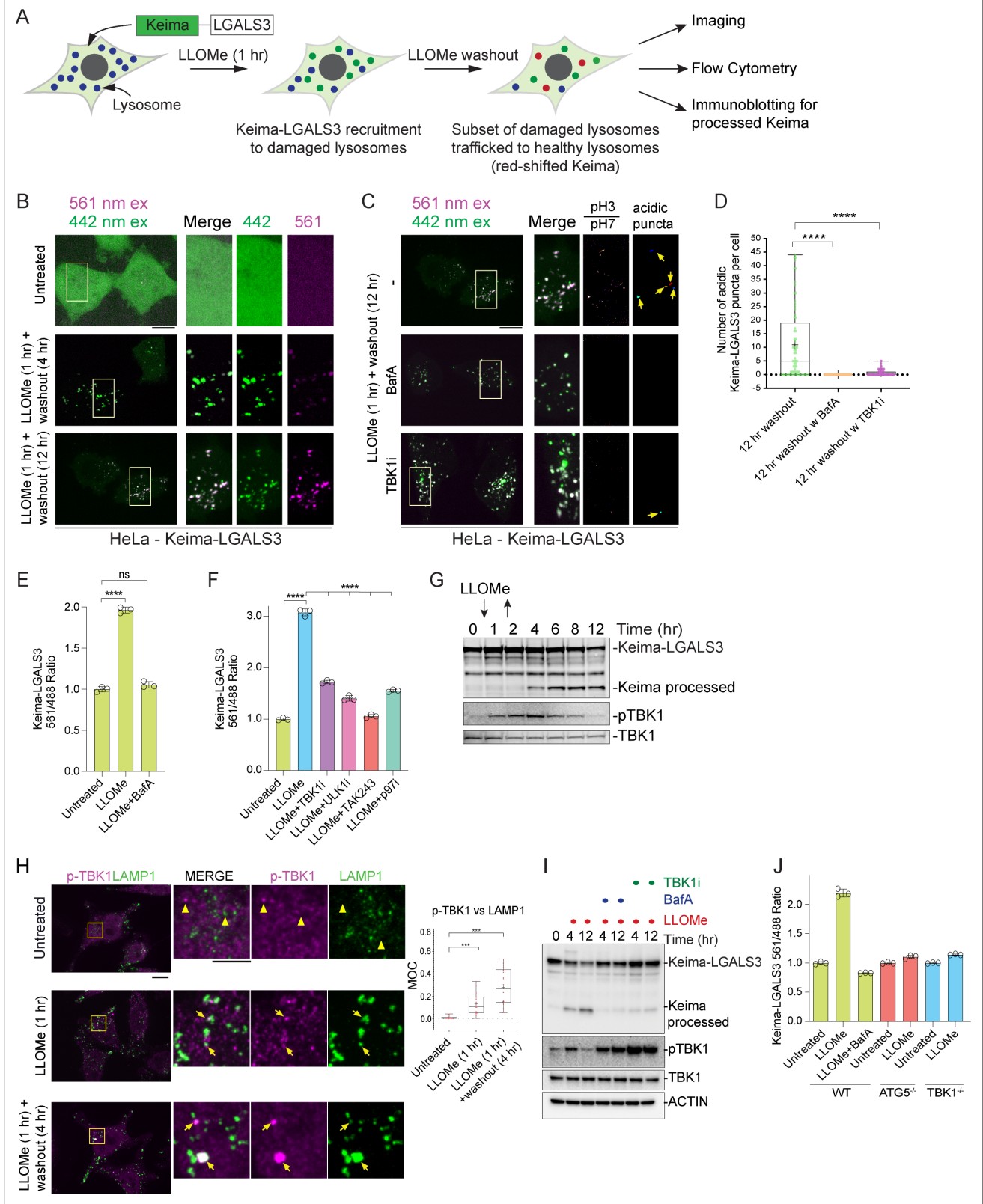

**Figure 5.** TBK1 is required for lysophagic flux. (**A**) Scheme depicting measurement of lysophagic flux using Lyso-Keima (Keima-LGALS3). Cells stably expressing Keima-LGALS3 are treated with LLOMe (1 hr), and the Keima-LGALS3 is recruited from the cytosol to damaged lysosomes, representing the initial recruitment step (green dot). After removing LLOMe (washout), damaged lysosomes undergo autophagy-dependent trafficking to a healthy lysosome, leading to a red-shift in Keima fluorescence (red dots) due to the acidic environment of the lysosome. Cells can be analyzed by imaging,

*Figure 5 continued on next page*

*Figure 5 continued*

flow cytometry or SDS–PAGE for processed Keima. (**B**) Keima-LGALS3 in untreated HeLa cells or in cells that were treated with LLOMe for 1 hr and the LLOMe washed out for 4 or 12 hr and imaged using excitation at 442 or 561 nm. Scale bar 10 μm. Zoom-in panels, 10 μm × 20 μm. (**C**) Keima-LGALS3 HeLa cells were either left untreated or treated for 1 hr followed by washout (12 hr) with or without prior addition of TBK1i or BafA. Cells were imaged using excitation at 442 or 561 nm. A ratio of the 561 nm/442 nm images was taken and puncta were identified from this 561 nm/442 nm image. Scale bar 10 μm. Zoom-in panels, 10 μm × 20 μm. (**D**) Quantification of Keima-positive lysosomes. 69 (untreated), 83 (BafA), and 66 (TBKi) cells were analyzed ****p < 0.0001. + marks the mean and the line is at the median. The plot represents merged data from three biological replicates. (**E**) Triplicate HeLa cells expressing Keima-LGALS3 were either left untreated or treated for 1 hr followed by washout (12 hr) with or without addition of BafA. Cells were then subjected to flow cytometry to measure the 561 nm/488 nm ratio. All values are normalized to the untreated sample. ****p < 0.0001. The plot represents mean and standard deviation from three biological replicates. (**F**) Triplicate HeLa cells expressing Keima-LGALS3 were either left untreated or treated for 1 hr followed by washout (12 hr) with or without prior addition of TBK1i, ULK1i, TAK243, and p97i. Cells were then subjected to flow cytometry to measure the 561 nm/488 nm ratio. All values are normalized to the untreated sample. ****p < 0.0001. The plot represents mean and standard deviation from three biological replicates. (**G**) HeLa cells expressing Keima-LGALS3 were either left untreated or treated for 1 hr followed by washout followed by harvesting at the indicated times. Lysed cells were then subjected to immunoblotting with the indicated antibodies. (**H**) Cells were left untreated (Untreated), treated with LLOMe for 1 hr and either fixed (LLOMe 1 hr) or the LLOMe was washed out for 4 hr prior to fixation (LLOMe 1 hr + washout 4 hr). Immunofluorescence was done using α-pTBK1/α-LAMP1 and imaging by confocal microscopy. Scale bar = 10 μm. Zoom-in panels, 10 μm × 10 μm. Right: quantification of localization using Mander's overlap coefficient (MOC). 23 (0 hr), 21 (1 hr), and 18 (4 hr washout) cells were analyzed for MOC. ***p < 0.001. + marks the mean and the line is at the median. The plot represents merged data from three biological replicates. (**I**) HeLa cells expressing Keima-LGALS3 were either left untreated or treated with LLOMe for 1 hr and then incubated for four or 12 hr post-washout in the presence or absence of either BafA or TBK1i. Cell lysates were subjected to immunoblotting using the indicated antibodies. (**J**) Triplicate WT, ATG5$^{-/-}$, or TBK1$^{-/-}$ HeLa cells expressing Keima-LGALS3 were either left untreated or treated for 1 hr followed by washout (4 hr) prior to flow cytometry to measure the 561 nm/488 nm ratio. All values are normalized to the untreated sample within each genotype. The plot represents mean and standard deviation from three biological replicates.

The online version of this article includes the following figure supplement(s) for figure 5:

**Source data 1.** Quantification of Keima-positive lysosomes.

**Source data 2.** 561/488 Keima ratios for *Figure 5E*.

**Source data 3.** 561/488 Keima ratios for *Figure 5F*.

**Source data 4.** Uncropped blots for *Figure 5G*.

**Source data 5.** MOC values for *Figure 5H*.

**Source data 6.** 561/488 Keima ratios for *Figure 5J*.

**Figure supplement 1.** Analysis of lysophagic flux using Lyso-Keima.

**Figure supplement 1—source data 1.** Uncropped blots.

## TBK1 is required for lysophagic flux

Given the recruitment of LGALS8 and Ub-binding autophagy receptors to damaged lysosomes and previous studies indicating that these proteins can bind and recruit TBK1 to autophagic cargo (*Thurston et al., 2009*; *Thurston et al., 2012*; *Thurston et al., 2016*), we explored TBK1 activation during lysosomal damage. First, as observed previously (*Nozawa et al., 2020*), we found that phosphorylation of TBK1 on Serine 172 (pS172, referred to as pTBK1) previously linked with activation of its kinase activity (*Kishore et al., 2002*) is evident after 1 hr of LLOMe and is maintained from 6-8 hr post-washout, returning to baseline by 12 hr in HeLa cells expressing the Keima-LGALS3 reporter (*Figure 5G*). Similarly, pTBK1 was detected on LGALS3-positive puncta after 1 hr LLOMe treatment and at 4 hr after 1 hr LLOMe treatment by immunofluorescence (mean MOC 0.1–0.3), indicating that the activated kinase is recruited to a subset of damaged lysosomes (*Figure 5H*). Thus, TBK1 activation and engagement of damaged lysosomes precede the earliest signs of lysophagic flux.

We next examined whether TBK1 was required for lysophagic flux. First, we found that a small molecule TBK1 inhibitor (MRT60821, referred to as TBK1i) added at the time of LLOMe washout, blocked Keima-LGALS3 flux by imaging analysis of acidic puncta (*Figure 5C–D*), by flow cytometry assays performed in biological triplicate (*Figure 5F*), and immunoblotting of processed Keima (*Figure 5I*) to an extent similar to that observed with BafA. Moreover, TBK1$^{-/-}$ HeLa cells were equally defective in Keima-LGALS3 flux as ATG5$^{-/-}$ cells by flow cytometry assays performed in biological triplicate, consistent with a major requirement for TBK1 in this process (*Figure 5J*). Interestingly, while pTBK1 was reduced to basal levels 12 hr post-LLOMe, pTBK1 remained fully elevated at this time in the presence of BafA or TBK1i (*Figure 5I*), suggesting that loss of pTBK1 could reflect autophagic degradation of the activated pool.

## Selectivity of Ub-binding autophagy receptors in lysophagic flux

Based on proteomic, immunofluorescence, and immunoblotting experiments described previously, multiple Ub- and TBK1-binding autophagy receptors (OPTN, CALCOCO2, and TAX1BP1) are rapidly recruited to damaged lysosomes (*Figure 4* and *Figure 4—figure supplement 1*). However, the contribution of the various autophagy receptors to actual lysophagic flux is unknown. To address this question, we first expressed Keima-LGALS3 in HeLa cells previously engineered to lack OPTN, CALCOCO2, and TAX1BP1 (referred to as triple knockout [TKO] cells; *Heo et al., 2015*) and measured lysophagic flux by flow cytometry in biological triplicate. The TKO mutant cells were as defective in LLOMe-stimulated lysophagic flux as cells lacking ATG5 (*Figure 6A*). In order to examine the extent to which each individual receptor is capable of promoting flux, we then reconstituted TKO cells expressing the Keima-LGALS3 reporters with EGFP-tagged OPTN, CALCOCO2, or TAX1BP1 (*Figure 6B and C*). While EGFP-TAX1BP1 and EGFP-OPTN rescued lysophagic flux, EGFP-CALCOCO2 was ineffective (*Figure 6B and C*). To further examine receptor specificity, we used gene editing to created mKeima-LGALS3 reporter cells lacking individual receptors (OPTN, TAX1BP1, CALCOCO2, or SQSTM1), as well as ATG7 as a control for canonical autophagy (*Figure 6—figure supplement 1A-E*). We found that cells lacking TAX1BP1 have the most severe block to lysophagic flux, phenotypically similar to cells lacking ATG7 examined in parallel (*Figure 6D*). Cells lacking SQSTM1 or OPTN had essentially wild-type lysophagic flux whereas CALCOCO2$^{-/-}$ cells had a partial reduction in lysophagic flux (*Figure 6D*). Consistent with a block in flux, TAX1BP1$^{-/-}$ cells – but not OPTN$^{-/-}$ or CALCOCO2$^{-/-}$ cells – displayed extensive LGALS3-positive puncta 10 hr post-washout after LLOMe treatment, as assessed using endogenous αLGALS3 staining by immunofluorescence (*Figure 6E*). The defect in TAX1BP1$^{-/-}$ cells was rescued by expression of EGFP-TAX1BP1, which also associated with LAMP1-positive puncta in LLOMe-treated cells, but expression of EGFP alone as a negative control failed to rescue clearance of, or associate with, damaged lysosomes (*Figure 6E*). Taken together, these data indicate that in HeLa cells, TAX1BP1 can drive lysophagic flux and that, among cargo receptors, TAX1BP1 is both necessary and sufficient for lysophagy. Moreover, overexpression of OPTN on its own can also promote lysophagy in cells lacking OPTN, TAX1BP1, and CALCOCO2, but is not required in HeLa cells. Interestingly, in our cell system, we were unable to validate the previous report based on RNAi that SQSTM1 is required for lysophagic flux (*Papadopoulos et al., 2017*).

## A system for quantitative analysis of lysophagic flux in iNeurons

Lysosomal function is linked with critical cellular functions during aging, and lysosomal dysfunction is linked with neurodegenerative diseases (*Peng et al., 2019*). As an initial step toward defining the mechanisms underlying removal of damaged lysosomes in neurons, we created a genetically tractable system for functional analysis of lysophagic flux. We employed a previously described hESC line that contains an inducible NGN2 gene, allowing for facile conversion to cortical-like iNeurons with >95% efficiency (*Ordureau et al., 2020*). We used PiggyBac transposase to create cells expressing RFP-EGFP-LGALS3 as a tandem reporter of lysophagic flux. Under basal conditions in 12- day iNeurons, the EGFP signal associated with RFP-EGFP-LGALS3 was largely localized in a diffuse cytosolic pattern, as expected (*Figure 7A*). These cells also contained RFP-positive puncta that colocalize with LAMP1-positive puncta (*Figure 7A*). These structures are indicative of either lysophagic flux occurring basally during the 12 -day differentiation process, nonselective bulk autophagy, or endocytosis of extracellular LGALS3 noted previously (*Furtak et al., 2001*; *Lepur et al., 2012*), as RFP is highly stable within the lysosome. To address these various possibilities, we mutated the carbohydrate recognition site Arginine 186 to Serine in LGALS3 (LGALS3$^{R186S}$) (*Aits et al., 2015*; *Delacour et al., 2007*) and monitored basal RFP-positive puncta in iNeurons. The expression of RFP-EGFP-LGALS3$^{R186S}$ did not result in significantly fewer RFP-positive puncta under basal conditions (*Figure 7—figure supplement 1C*), indicating LGALS3 translocation into the lysosome was not due to an increase in damaged lysosomes and likely either represents nonselective bulk autophagic flux or increased endocytosis.

In contrast to basal, untreated conditions, iNeurons treated with LLOMe (1 hr) displayed an increase in the number of EGFP-positive puncta per cell (*Figure 7B and C*). However, 12 hr post-washout, the EGFP-positive puncta associated with the tandem LGALS3 reporter had been cleared, and the number of puncta returned to near basal conditions (*Figure 7B and C*). Importantly, the clearing of EGFP-positive puncta was largely blocked by BafA and VPS34i, as expected if the EGFP-positive puncta were cleared via lysophagy (*Figure 7D and E*). Moreover, as expected, the LGALS3$^{R186S}$ mutant failed

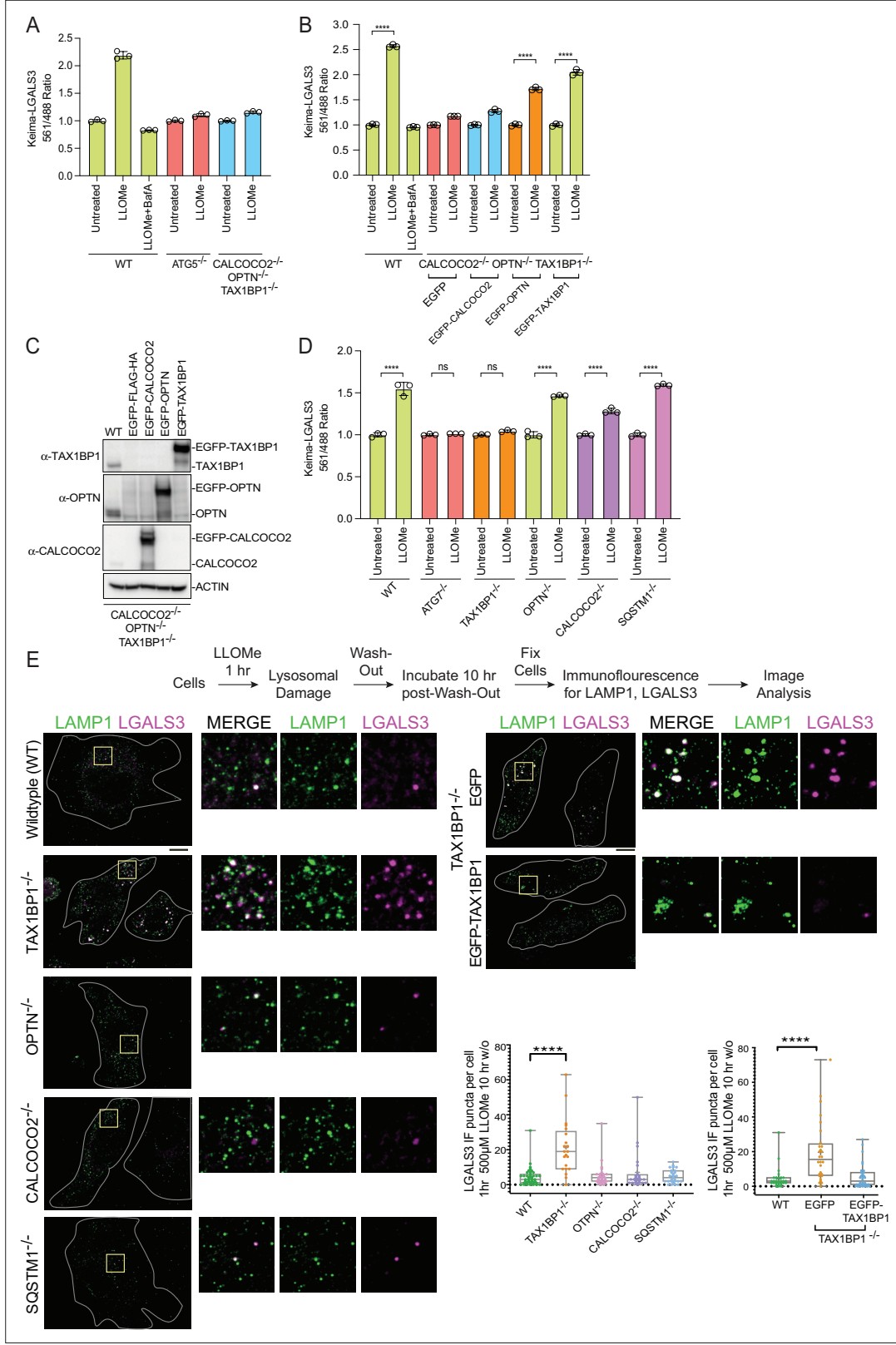

**Figure 6.** Role for Ub-binding autophagy receptors in lysophagy. (**A**) Triplicate WT; ATG5[−/−]; or OPTN[−/−];
TAX1BP1[−/−]; CALCOCO2[−/−] (TKO) HeLa cells expressing Keima-LGALS3 were either left untreated or treated
for 1 hr followed by washout (12 hr) prior to flow cytometry to measure the 561 nm/488 nm ratio. All values are
normalized to the untreated sample within each genotype. The plot represents mean and standard deviation from

*Figure 6 continued on next page*

*Figure 6 continued*

three biological replicates. (**B**) Triplicate WT or TKO HeLa cells expressing Keima-LGALS3 were reconstituted with lentivirally expressed EGFP-FLAG-HA, EGFP-CALCOCO2, EGFP-OPTN, or EGFP-TAX1BP1. Cells were either left untreated or treated for 1 hr followed by washout (12 hr) prior to flow cytometry to measure the 561 nm/488 nm ratio. As a control for lysophagic flux, some samples were also treated with BafA during the washout. All values are normalized to the untreated sample within each genotype. ****p < 0.0001. The plot represents mean and standard deviation from three biological replicates. (**C**) Cells from panel B were lysed and subjected to immunoblotting with the indicated antibodies. (**D**) HeLa cells expressing Keima-LGALS3 (with or without deletion of ATG7, TAX1BP1, OPTN, CALCOCO2, or SQSTM1) were either left untreated or treated for 1 hr followed by washout (12 hr) prior to flow cytometry to measure the 561 nm/488 nm ratio. All values are normalized to the untreated sample within each genotype. ****p < 0.0001. The plot represents mean and standard deviation from three biological replicates. (**E**) HeLa cells (with or without deletion of TAX1BP1, OPTN, CALCOCO2, or SQSTM1) were either left untreated or treated for 1 hr followed by washout (10 hr) prior to immunostaining with α-LAMP1 (green) and α-LGALS3 (magenta). The number of LGALS3 puncta per cell present after washout is plotted (right top panel). The block to lysophagic flux was rescued by expression of EGFP-TAX1BP1 but not EGFP (lower right panel). 41 (WT), 21 (TAX1BP1), 25 (OPTN), 21 (CALCOCO2), and 27 (SQSTM1) cells were analyzed in the upper graph. 29 (WT), 28 (EGFP), and 32 (EGFP-TAX1BP1) cells were analyzed in the bottom graph. ****p < 0.0001. Scale bar 10 µm. Zoom-in panels, 10 µm × 10 µm. + marks the mean and the line is at the median. The plot represents merged data from three biological replicates.

The online version of this article includes the following figure supplement(s) for figure 6:

**Source data 1.** 561/488 Keima ratios for *Figure 6A*.

**Source data 2.** 561/488 Keima ratios for *Figure 6B*.

**Source data 3.** 561/488 Keima ratios for *Figure 6D*.

**Source data 4.** The number of galectin puncta per cell post-washout for *Figure 6E*.

**Source data 5.** Uncropped blots for *Figure 6C*.

**Figure supplement 1.** Characterization of Ub-cargo receptor mutant cell lines.

**Figure supplement 1—source data 1.** Uncropped blots.

to be recruited to damaged lysosomes (*Figure 7—figure supplement 1A-C*). These data indicate that the clearing of EGFP-positive puncta can be used as a means by which to examine lysophagic flux in iNeurons, as previously demonstrated in HeLa cells (*Maejima et al., 2013*).

## Lysophagic flux in iNeurons requires TAX1BP1 and TBK1

In order to examine the TBK1-cargo receptor axis in the iNeuron system, we initially employed the TBK1i small molecule inhibitor during a 12 hr washout after a 1 hr LLOMe treatment. TBK1i blocked LGALS3 clearance to an extent comparable to that observed with BafA, indicating that lysophagic flux in iNeurons requires TBK1 activity (*Figure 7D and E*). We next employed gene editing to create ES:NGN2:LGALS3 tandem reporter cells lacking TAX1BP1 (*Figure 7—figure supplement 1D,E*). Deletion of TAX1BP1 led to substantial reduction in clearance of EGFP-positive puncta during a 12 hr washout after LLOMe (1 hr) treatment (*Figure 7F and G*). Additional deletion of OPTN with TAX1BP1 did not further exacerbate clearance of EGFP-positive puncta at 12 hr washout after LLOMe (1 hr) treatment (*Figure 7F and G*, and *Figure 7—figure supplement 1F,G*). Thus, these data indicate that TAX1BP1 and TBK1 collaborate to promote facile clearance of damaged lysosomes in iNeurons.

## Role for TAX1BP1 SKICH domain in lysophagy

Ub-binding cargo receptors typically contain three major structural elements: coiled-coil (CC) motifs, LIR motifs that bind to ATG8 proteins, and C-terminal Ub-binding domains, which include UBAN and ZnF domains (*Johansen and Lamark, 2020*; *Figure 8A and B*). In addition, TAX1BP1 also contains an N-terminal SKICH domain, which interacts with the TBK1-binding adaptor protein NAP1, to facilitate TBK1 binding (*Fu et al., 2018*). Interestingly, TAX1BP1 has also been recently shown to bind RB1CC1—a component of the ULK1 kinase complex required for autophagy—in a manner that requires Alanine 114 within the SKICH domain and a LGALS8-binding element located between residues 632 and 639 (*Bell et al., 2021*; *Ohnstad et al., 2020*; *Figure 8A*). In order to probe the activities of these functional elements in lysophagy, we stably expressed various TAX1BP1 mutants in HeLa TKO cells and measured lysophagic flux by flow cytometry in biological triplicate after lysosomal damage

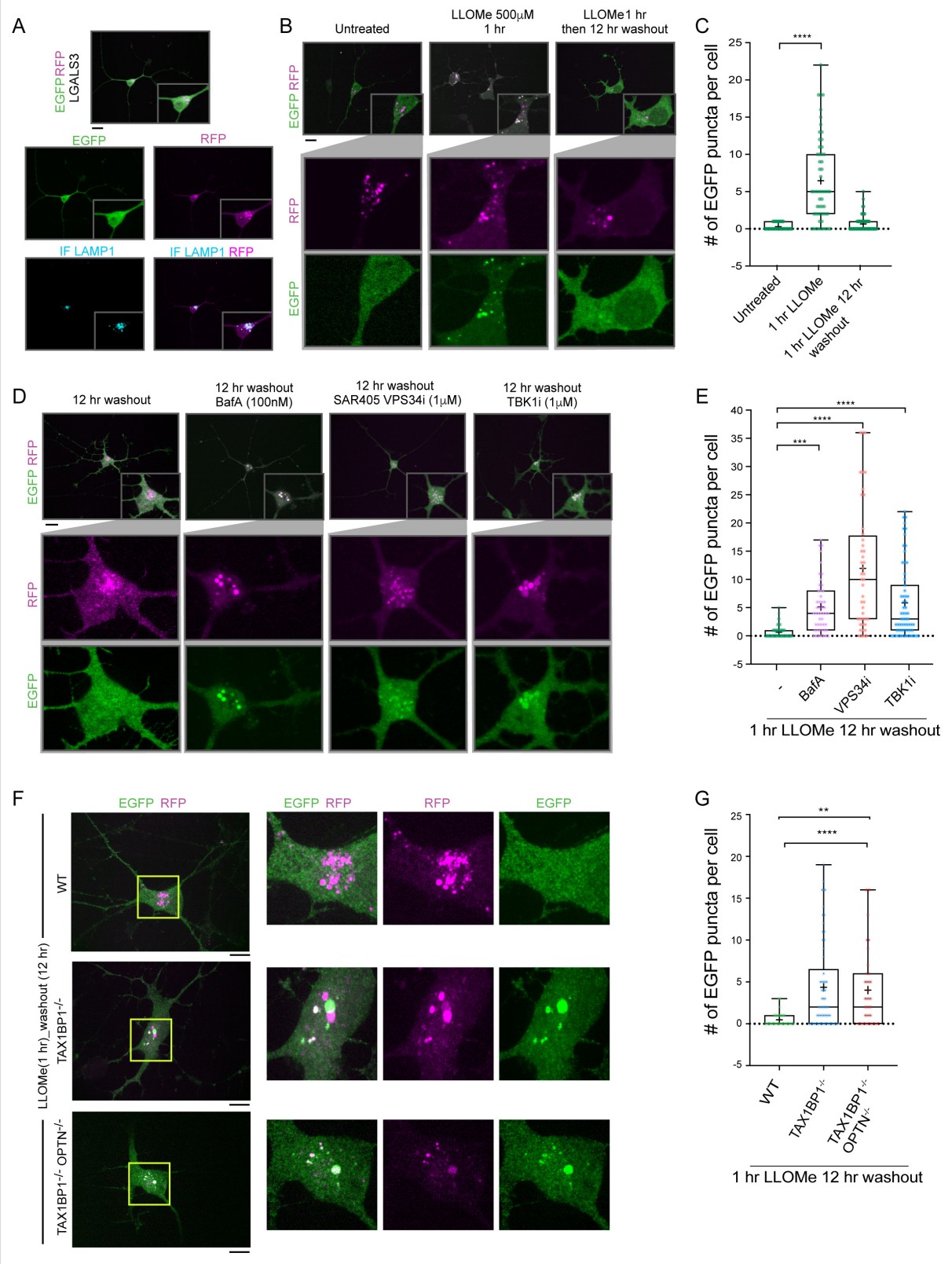

**Figure 7.** TAX1BP1 promotes lysophagic flux in iNeurons. (**A**) RFP-EGFP-LGALS3 is trafficked to lysosomes in iNeurons. ES cells expressing RFP-EGFP-LGALS3 via a PiggyBac vector were converted to iNeurons using inducible NGN2 (see Materials and methods) and imaged for EGFP, RFP, and LAMP1 using α-LAMP1 antibodies. While EGFP signal was diffusely localized in the soma, RFP-positive puncta colocalized with lysosomes based on colocalization with LAMP1 staining, indicating that a subset of the RFP-EGFP-LGALS3 protein is trafficked to the lysosome under basal conditions.

*Figure 7 continued on next page*

*Figure 7 continued*

Scale bar = 20 µm. iN soma zoom-in panels, 30 µm × 40 µm. (**B**) iNeurons expressing RFP-EGFP-LGALS3 were either left untreated, treated with LLOMe for 1 hr, or treated with LLOMe for 1 hr followed by a 12 hr washout. Cells were imaged for EGFP and RFP and the number of EGFP puncta per cell quantified. Loss of EGFP puncta during the washout period is indicative of lysophagic flux. Scale bar = 20 µm. (**C**) Quantification of EGFP puncta per cell after washout from experiments in panel B. The average EGFP puncta per cell was 0.289 at 0 hr (45 cells), 6.46 at 1 hr LLOMe (55 cells) and 0.652 at 12 hr washout after LLOMe (66 cells). ****p < 0.0001. + marks the mean and the line is at the median. The plot represents merged data from three biological replicates. (**D**) iNeurons were subjected to LLOMe treatment and washout as in panel B but treated with or without TBK1i, VPS34i, or BafA during the washout period. Cells were imaged for EGFP and RFP. Scale bar = 20 µm. iN soma zoom-in panels, 30 µm × 40 µm. (**E**) Quantification of EGFP puncta per cell from the experiment in panel D. The average EGFP puncta per cell at 12 hr washout was 0.65 with no inhibitor (66 cells), 5.14 with BafA (49 cells), 11.95 with VPS34i (44 cells), and 5.84 with TBKi (63 cells). ****p < 0.0001, ***p < 0.001. + marks the mean and the line is at the median. The plot represents merged data from three biological replicates. (**F**) WT, TAX1BP1⁻/⁻, or TAX1BP1⁻/⁻; OPTN⁻/⁻ iNeurons were subjected to LLOMe treatment and washout as in panel B. Cells were imaged for EGFP and RFP. Scale bar = 10 µm. iN soma zoom-in panels, 20 µm × 20 µm. (**G**) Quantification of EGFP puncta per cell from the experiment in panel F. The average EGFP puncta per cell at 12 hr washout after LLOMe for wild-type cells was 0.474 (38 cells), for TAX1BP1⁻/⁻ cells was 4.36 (62 cells) and for TAX1BP1⁻/⁻; OPTN⁻/⁻ cells was 4.03 (39 cells). ****p < 0.0001, **p< 0.01. + marks the mean and the line is at the median. The plot represents merged data from three biological replicates.

The online version of this article includes the following figure supplement(s) for figure 7:

**Source data 1.** The number of GFP puncta per cell for *Figure 7C*.

**Source data 2.** The number of GFP puncta per cell for *Figure 7E*.

**Source data 3.** The number of GFP puncta per cell for *Figure 7G*.

**Figure supplement 1.** Analysis of LGALS3ᴿ¹⁸⁶ˢ recruitment to damaged lysosomes in iNeurons.

**Figure supplement 1—source data 1.** The number of EGFP puncta per cell for *Figure 7—figure supplement 1B and C*.

**Figure supplement 1—source data 2.** Uncropped blots for *Figure 7—figure supplement 1E, G*.

with LLOMe (*Figure 8C and D*). Consistent with a recent report (*Ohnstad et al., 2020*), we found that this collection of TAX1BP1 mutants displayed differential levels of expression, with several mutants including 632–639Δ displaying elevated levels compared to WT TAX1BP1 (*Figure 8D*). In addition, although the antibody employed does not react with the CC2Δ mutant, we demonstrated that the expression level of this mutant is equivalent to that of CC1Δ and CC3Δ based on EGFP fluorescence measured by flow cytometry (*Figure 8—figure supplement 1A*). We found that various TAX1BP1 mutations display varying levels of activity. First, deletion of the SKICH domain (1–140Δ) resulted in the complete inhibition of lysophagic flux, comparable to that observed in the TKO mutant, despite being expressed at a level higher than WT (*Figure 8C and D*). Consistent with this, expression of the A114Q mutant also severely blocked lysophagic flux. In contrast, mutations that affect LGALS8 binding (Y635A, N637A, and 632–639Δ) or removal of any of the CC domains had little or no impact on lysophagic flux (*Figure 8C*). The V192S mutation that affects binding to ATG8 proteins reduced activity by ~40 % (*Figure 8C*). Finally, mutation of TAX1BP1's C-terminal ZnF domain resulted in a partial (~50%) reduction in activity (*Figure 8C*), but still retained the ability to be recruited to damaged lysosomes (*Figure 8—figure supplement 1B*). However, previous studies have shown that this mutation has residual Ub-binding activity (*Tumbarello et al., 2015*), potentially accounting for residual activity. Therefore, to examine a direct role for Ub in TAX1BP1 recruitment, we treated cells with E1i before LLOMe treatment and during a 4 hr washout (*Figure 8—figure supplement 1C*). We found that inhibition of Ub conjugation completely blocked TAX1BP1 recruitment to damaged lysosomes, while TAX1BP1 recruitment was unaffected by treatment with p97i or TBK1i (*Figure 8—figure supplement 1C*). In parallel, the D474N Ub-binding mutant of OPTN was tested and did not rescue the TKO phenotype to promote lysophagy while the TBK1 phosphoresistant mutants of OPTN expressed in the TKO background did rescue lysophagic flux (*Heo et al., 2015*; *Figure 8E and F*). Moreover, OPTNᴰ⁴⁷⁴ᴺ failed to be recruited to damaged lysosomes, as assessed by microscopy (*Figure 8—figure supplement 1B*). Taken together, these data indicate that recruitment of both TAX1BP1 and OPTN to damaged lysosomes requires upstream ubiquitylation.

TAX1BP1 activity appears to depend extensively on its N-terminal SKICH domain, which associates with TBK1-NAP1. When assessed by fluorescence microscopy, the A114Q and the SKICH domain mutants appeared to localize to the same region as damaged lysosomes (*Figure 8—figure supplement 1B*). Although these mutants are able to translocate to the lysosomal region, we found that LLOMe-dependent phosphorylation of TBK1 in TKO cells reconstituted with WT and mutant TAX1BP1 required A114Q and the SKICH domain (*Figure 8G*), which correlates with the loss of lysophagic

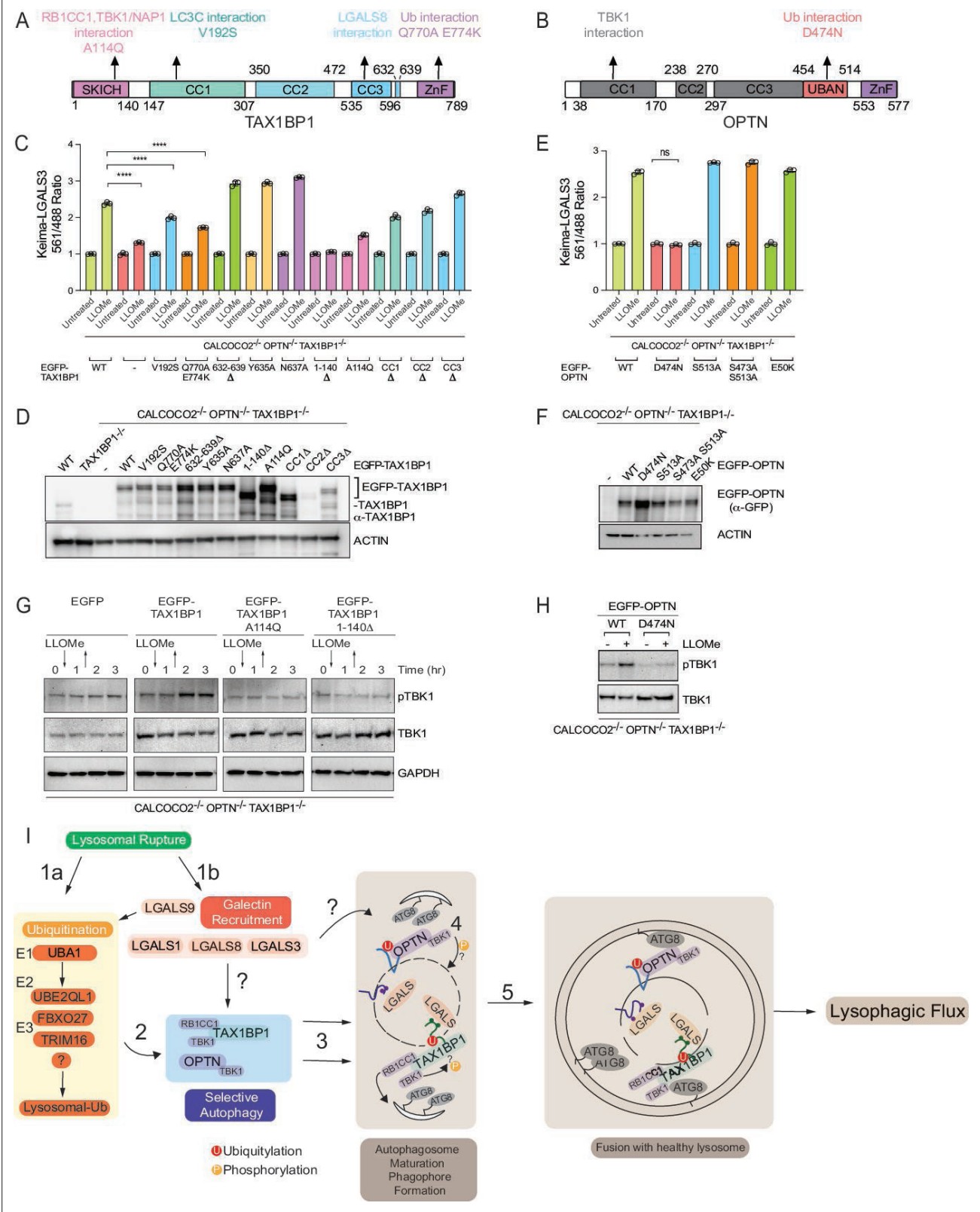

**Figure 8.** Structure–function analysis of TAX1BP1 and OPTN for lysophagy. (**A**) Domain structure of TAX1BP1 showing the location of mutations examined in this study. (**B**) Domain structure of OPTN showing the location of mutations examined in this study. (**C**) HeLa TKO cells expressing Keima-LGALS3 were infected with lentiviruses expressing GFP-tagged WT or mutant TAX1BP1 proteins to obtain stable expression. Cells in biological triplicate were either left untreated or treated for 1 hr followed by washout (12 hr) prior to flow cytometry to measure the 561 nm/488 nm ratio. All

*Figure 8 continued on next page*

*Figure 8 continued*

values are normalized to the untreated sample within each genotype. The plot represents mean and standard deviation from three biological replicates. ****p < 0.0001 (**D**) Immunoblot of cell extracts from panel C probed with α-TAX1BP1 or α-actin as a loading control. Note that some mutants are highly stabilized, as reported previously (***Ohnstad et al., 2020***). The EGFP-TAX1BP1 CC2Δ mutant is not detected by western blot due to the loss of the epitope-binding site of the antibody, nevertheless is detected by FACS (***Figure 8—figure supplement 1A***). (**E**) HeLa TKO cells expressing Keima-LGALS3 were infected with lentiviruses expressing GFP-tagged WT or mutant OPTN proteins to obtain stable expression. Cells in biological triplicate were either left untreated or treated for 1 hr followed by washout (12 hr) prior to flow cytometry to measure the 561 nm/488 nm ratio. The plot represents mean and standard deviation from three biological replicates. ****p < 0.0001. (**F**) Immunoblot of cell extracts from panel E probed with α-GFP or α-actin as a loading control. (**G**) HeLa TKO cells were infected with lentiviruses expressing EGFP-tagged WT or mutant TAX1BP1 proteins to obtain stable expression. Cells were either left untreated or treated for 1 hr with LLOMe followed by washout. Cells were harvested at the indicated times and subjected to immunoblotting with the indicated antibodies. (**H**) HeLa TKO cells were infected with lentiviruses expressing EGFP-tagged WT or mutant OPTN proteins to obtain stable expression. Cells were either left untreated or treated for 1 hr with LLOMe followed by washout (12 hr). Cells were harvested at the indicated times and subjected to immunoblotting with the indicated antibodies. (**I**) Model figure. Lysosomal rupture leads to the parallel recruitment of galectins and unleashes a wave of ubiquitination on the lysosome (Steps 1 a and b). In step 2, ubiquitination promotes the recruitment of both OPTN-TBK1 and TAX1BP1-TBK1-RB1CC1 complexes to the damage lysosome, thereby promoting de novo phagophore formation and local TBK1 activation to drive efficient lysophagy (Steps 3–5).

The online version of this article includes the following figure supplement(s) for figure 8:

**Source data 1.** 561/488 Keima ratios for *Figure 8C*.

**Source data 2.** 561/488 Keima ratios for *Figure 8E*.

**Source data 3.** Uncropped blots for *Figure 8*.

**Figure supplement 1.** Structure–function analysis of TAX1BP1 and OPTN for lysophagy.

**Figure supplement 1—source data 1.** The number of colocalized TAX1BP1- and LAMP1-positive puncta.

flux with these mutations, and is also consistent with the genetic requirement for TBK1 in lysophagic flux. As expected (***Heo et al., 2015***), TBK1 activation in the context of OPTN-mediated lysophagy in TKO cells was absolutely dependent upon the ability of OPTN to bind Ub, as the D747N mutant was unable to support TBK1 phosphorylation upon LLOMe treatment (***Figure 8H***).

## Discussion

The lysosome is the terminal degradative organelle for the autophagic and endocytic pathways, and as a membrane-bound organelle itself, is also susceptible to damage from a plethora of sources. Irrevocably damaged lysosomes are eliminated by the selective autophagic pathway of lysophagy, which requires: (1) galectin binding to damaged lysosomes, (2) ubiquitination of lysosomal components, (3) core components of the autophagic machinery such as the VPS34 kinase complex, the ULK1-RB1CC1-ATG13 module, and the ATG5-ATG12 lipidation cascade, and (4) the biogenesis of newly formed lysosomes via the TFEB transcriptional response (***Nakamura et al., 2020***). Despite the identification of these individual steps in the lysophagy pathway, many events such as the role of Ub-binding cargo receptors remain poorly characterized. Using a suite of quantitative proteomic techniques, we have generated a landscape of the damaged lysosome and have identified key regulatory modules in the pathway. Lysosomal damage leads to the sequential recruitment of ESCRT-III complex proteins, galectin proteins LGALS1, 3, and 8, and ATG8 family of proteins, consistent with prior observations (***Jia et al., 2018***). Additionally, the proteomic data provided here provide a variety of candidates for future hypothesis-driven investigations into the molecular mechanisms of lysophagy, including the development of a spatial understanding of distinct galectins and their recruitment to damaged lysosomes.

Our proximity biotinylation maps of the ATG8 orthologs MAP1LC3B and GABARAPL2 in response to LLOMe revealed rapid, specific biotinylation of proteins associated with the lysosomal membrane, as compared to other organelles, consistent with the selectivity of this response. Interestingly, both MAP1LC3B and GABARAPL2 utilize their LIR docking sites to recruit a subset of downstream factors to the damaged lysosome. Proximity biotinylation maps of LGALS1, 3, and 8 also revealed galectin-specific interactions with LGALS8 having apparently selective interactions with the MTOR signaling machinery and autophagy. These results are consistent with prior observations that LGALS8 regulates the autophagic response after lysosomal damage and suggest that LGALS8 is a key regulatory node for lysophagy (***Jia et al., 2018***).

Prior observations have indicated that the selective autophagy receptors, SQSTM1 and TAX1BP1, are recruited to damaged lysosomes, and SQSTM1 has been reported to be required for lysophagy in HeLa cells using siRNA (*Papadopoulos et al., 2017*). Our proteomics data clearly demonstrate that SQSTM1, CALCOCO2, OPTN, and TAX1BP1 are recruited to the lysosome within 30 min post-damage, raising questions about the actual identity of the relevant receptors. Are all recruited receptors necessary for lysophagy, perhaps playing distinct roles or are some receptors recruited but not required? To address this question, we developed a lysophagic flux assay—Lyso-Keima—and demonstrated in HeLa cells that deletion of TAX1BP1 was sufficient to eliminate lysophagic flux. Reduced lysophagic flux was also found in iNeurons lacking TAX1BP1. In HeLa cells lacking TAX1BP1, OPTN, and CALCOCO2, TAX1BP1 and to a lesser extent OPTN, but not CALCOCO2, can rescue lysophagy. In iNeurons, the phenotype of TAX1BP1$^{-/-}$; OPTN$^{-/-}$ double mutants was similar to TAX1BP1 deletion alone, suggesting that OPTN does not play a supporting role in this process in this context. The dramatic reduction in lysophagic flux in HeLa cells or iNeurons solely lacking TAX1BP1 also indicates that SQSTM1 is not sufficient to support flux in these cells under the conditions employed here. These results, together with the results of prior studies on other types of selective autophagy, indicate distinct receptor requirements for individual cargo. For example, in HeLa cells, Parkin-dependent mitophagic flux requires primarily OPTN and CALCOCO2 (*Wong and Holzbaur, 2014*; *Heo et al., 2015*; *Lazarou et al., 2015*), in addition to TBK1 and xenophagy require CALCOCO2 (*Ravenhill et al., 2019*). Moreover, our results indicate that the apparent strength of receptor recruitment to damaged organelles is not a surrogate for a functional role.

Overall, our data support the model in *Figure 8I*. Lysosome rupture leads to two apparently independent pathways, one resulting in the ubiquitylation of lysosomal proteins and the other reflecting association of galectins with glycosylated luminal domains in lysosomal membrane proteins. Recruitment of LGALS3 or LGALS8 does not require the Ub pathway downstream of UBE2QL1 (*Koerver et al., 2019*), and likewise, the activation of the ubiquitylation arm of the pathway does not require LGALS3 or LGALS8 (*Jia et al., 2020a*) However, it has been reported that depletion of LGALS9 leads to reduced lysosomal ubiquitylation indicating some level of cross-talk between the two pathways (*Jia et al., 2020a*). Precisely which ubiquitylation systems are involved and how they mechanistically are linked with LGALS9 remains unknown. Our results indicate that the Ub arm of the pathway is critical for recruitment of TAX1BP1 and OPTN. First, inhibition of the UBA1 E1-activating enzyme blocks TAX1BP1 recruitment to damaged lysosomes and point mutatations in TAX1BP1's C-terminal ZnF domain that reduce, but do not eliminate Ub-binding, displayed reduced lysophagic flux in response to LLOMe. Interestingly, inhibition of p97 or TBK1 did not block TAX1BP1 recruitment to damaged lysosomes, although both inhibitors blocked flux and therefore p97 and TBK1 presumably have roles downstream of TAX1BP1 recruitment. Second, mutation of OPTN's Ub-binding UBAN domain abolishes recruitment to damaged lysosomes and blocks flux in the context of rescue of the TAX1BP1/OPTN/CALCOCO2 triple mutant HeLa cells. Given that TAX1BP1 can also interact with overexpressed LGALS8 independent of lysosomal damage (*Bell et al., 2021*; *Huttlin et al., 2021*), it is formally possible that recruitment can occur via both Ub-dependent and -independent pathways under some conditions.

Both TAX1BP1 and OPTN associate with the TBK1 protein kinase, albeit through distinct structural motifs. Several findings link TBK1 with lysophagy. First, small molecular inhibitors of TBK1 block lysophagic flux in HeLa cells and iNeurons, and deletion of TBK1 blocks lysophagy in HeLa cells. Second, deletion of the SKICH domain of TAX1BP1 or mutation of A114 which is required for TBK1–NAP1 association results in loss of activity in lysophagic flux assays. Third, while reintroduction of WT TAX1BP1 into TAX1BP1$^{-/-}$ HeLa cells activates TBK1 phosphorylation in response to LLOMe, removal or mutation of the SKICH domain abolishes TBK1 activation in TKO cells, indicating that TBK1 activation depends on its association with TAX1BP1. The targets of TBK1 in this context remain to be identified, but we note that the cargo adaptors themselves are substrates of TBK1 in other types of selective autophagy (*Heo et al., 2015*; *Richter et al., 2016*), and RAB7 is a substrate of TBK1 in response to signals that induce mitophagy (*Heo et al., 2018*).

The observation that TAX1BP1 is a major receptor for lysophagy extends recent work identifying roles for the protein in various selective autophagic pathways (*Gubas and Dikic, 2021*). TAX1BP1 likely plays a dual role, as it interacts with both TBK1 and the RB1CC1–ULK1–ATG13 complex through its SKICH domain (*Figure 8I*). This may allow TAX1BP1 to orchestrate signaling via both of these

kinase complexes, and could possibly promote autophagosome formation directly via recruitment of ULK1 to cargo (*Ravenhill et al., 2019*; *Shi et al., 2020*; *Turco et al., 2020*; *Vargas et al., 2019*). In addition, TAX1BP1 appears to have diverse cargo, ranging from membranous organelles as shown here to ubiquitylated protein aggregates as recently described (*Sarraf et al., 2020*). TBK1, OPTN, p97, and other factors involved in selective autophagy are mutated in a variety of neurological disorders such as ALS and FTD (*Cirulli et al., 2015*; *Freischmidt et al., 2015*). Further elucidation of the mechanisms used by these proteins and their relationship to other components such as TAX1BP1 will assist in the development of therapeutic approaches.

# Materials and methods

**Key resources table**

| Reagent type (species) or resource | Designation | Source or reference | Identifiers | Additional information |
|---|---|---|---|---|
| Cell line (*Homo sapiens*) | Hela Flp-in-TRex | This paper | | Obtained from Brian Raught, Ontario Cancer institute |
| Cell line (*Homo-sapiens*) | Hela | ATCC | CCL-2; RRID:CVCL_0030 | |
| Cell line (*Homo-sapiens*) | HEK293T | ATCC | CRL-1573; RRID:CVCL_0045 | |
| Cell line (*Homo-sapiens*) | H9 | Wicell | WA9, CVCL_9773; RRID: CVCL_9773 | |
| Antibody | Galectin-1/LGALS1 (D608T) (Rabbit mAb, monoclonal) | Cell Signaling Technology | #12936 S; RRID:AB_2137707 | IF (1:300), WB (1:1000) |
| Antibody | Galectin-3/ LGALS3 (Rabbit Antibody, polyclonal) | Proteintech | 60207–1-I; RRID:AB_10951109 | IF (1:300), WB (1:1000) |
| Antibody | Galectin-3/ LGALS3 Antibody (M3/38) for immunofluorescence (rat, monoclonal) | Santa-Cruz | sc-23938; RRID:AB_627658 | IF (1:300), WB (1:1000) |
| Antibody | Human Galectin-8/LGALS8 Antibody, (goat polyclonal) | R&D Systems | AF1305; RRID:AB_2137229 | IF (1:300), WB (1:1000) |
| Antibody | LC3B D11 (Rabbit mAb, monoclonal) | Cell Signaling Technology | #3868 S; RRID:AB_2137707 | IF (1:300), WB (1:1000) |
| Antibody | GABARAPL2 (D1W9T) (Rabbit mAb, monoclonal) | Cell Signaling Technology | 14256; RRID:AB_2798436 | IF (1:300), WB (1:1000) |
| Antibody | Anti-CALCOCO2 antibody, (Rabbit, polyclonal) | Abcam | ab68588; RRID:AB_1640255 | IF (1:300), WB (1:1000) |
| Antibody | Anti-OPTN (rabbit, polyclonal) | Sigma | HPA003279; RRID:AB_1079527 | IF (1:300), WB (1:1000) |
| Antibody | Anti-TAX1BP1 (rabbit, polyclonal) | Sigma | HPA024432; RRID:AB_1857783 | IF (1:300), WB (1:1000) |
| Antibody | SQSTM1 monoclonal antibody (M01), clone 2C11 (mouse, monoclonal) | Abnova | H00008878-M01; RRID: AB_437085 | IF (1:300), WB (1:1000) |
| Antibody | Raptor (24C12) (Rabbit, monoclonal) | Cell Signaling Technology | #2280 S; RRID:AB_561245 | IF (1:300), WB (1:1000) |
| Antibody | mTOR (7C10) (Rabbit, monoclonal) | Cell Signaling Technology | #2983; RRID:AB_2105622 | IF (1:300), WB (1:1000) |
| Antibody | NPC1 (Rabbit, monoclonal) | Abcam | ab134113; RRID: AB_2734695 | IF (1:300), WB (1:1000) |
| Antibody | LAMP1 (D2D11) Rabbit (Rabbit, monoclonal) | Cell Signaling Technology | #9091 S; RRID:AB_2687579 | IF (1:300), WB (1:1000) |

*Continued on next page*

*Continued*

| Reagent type (species) or resource | Designation | Source or reference | Identifiers | Additional information |
|---|---|---|---|---|
| Antibody | LAMP1 (D401S) (Mouse, monoclonal) | Cell Signaling Technology | #15665 S; RRID: AB_2798750 | IF (1:300), WB (1:1000) |
| Antibody | Anti-TMEM192 antibody [EPR14330] (Rabbit, monoclonal) | Abcam | ab185545, Discontinued | IF (1:300), WB (1:1000) |
| Antibody | Anti-HA, (Mouse, monoclonal) | Biolegend | #901513; RRID:AB_2565335 | IF (1:300), WB (1:1000) |
| Antibody | Anti-Flag M2 (mouse, monoclonal) | Sigma | F1804; RRID: AB_262044 | IF (1:300), WB (1:1000) |
| Antibody | Anti-Keima-Red (Mouse, monoclonal) | MBL international | M182-3M; RRID: AB_10794910 | IF (1:300), WB (1:1000) |
| Antibody | phospho-TBK1/NAK (Ser172) (D52C2) (Rabbit, monoclonal) | Cell Signaling Technology | #5483 S; RRID:AB_10693472 | IF (1:300), WB (1:1000) |
| Antibody | TBK1/NAK (Rabbit, polyclonal) | Cell Signaling Technology | #3013 S; RRID: AB_2199749 | IF (1:300), WB (1:1000) |
| Antibody | beta-actin (mouse, monoclonal) (AC-15) | Santa Cruz | sc-69879; RRID: AB_1119529 | IF (1:300), WB (1:1000) |
| Antibody | Anti-GFP (Mouse, monoclonal) | Roche | #11814460001; RRID:AB_390913 | IF (1:300), WB (1:1000) |
| Strain, strain background (*Escherichia coli*) | DH5 alpha *E. coli* competent cells | Homemade | | |
| Strain, strain background (*E. coli*) | T1R *E. coli* Competent cells | Homemade | | |
| Chemical compound, drug | Gly-Phe-β-naphthylamide | Cayman Chemical | #14634 | |
| Chemical compound, drug | L-Leucyl-L-Leucine methyl ester (hydrochloride) | Cayman Chemical | #16008 | |
| Chemical compound, drug | Biotin Tyramide | Iris Biotech(peptide solutions) | LS-3500.0250 | |
| Chemical compound, drug | Trolox | Cayman Chemical | #53188-07-1 | |
| Chemical compound, drug | Hydrogen peroxide solution | Sigma | H1009 | |
| Chemical compound, drug | Pierce Anti-HA Magnetic Beads | Thermo Scientific | #88837 | |
| Chemical compound, drug | TMTpro 16-plex Label Reagent Set | Thermo Scientific | A44520 | |
| Chemical compound, drug | IKKε/TBK1 Inhibitor II, MRT67307 | EMD millipore | CAS 1190378-57-4 | |
| Chemical compound, drug | ULK1 inhibitor, MRT68921 | Cayman chemical | #1190379-70-4 | |
| Chemical compound, drug | TAK-243 | SelleckChem | S8341 | |
| Chemical compound, drug | CB-5083 | Cayman Chemical | S810 | |
| Chemical compound, drug | Bafilomycin A1 | Cayman Chemical | #88899-55-2 | |

*Continued*

| Reagent type (species) or resource | Designation | Source or reference | Identifiers | Additional information |
|---|---|---|---|---|
| Commercial assay or kit | Lipofectamine 3,000 | Invitrogen | L3000008 | |
| Commercial assay or kit | Pierce High pH Reversed-Phase Peptide Fractionation Kit | ThermoFisher Scientific | #84868 | |
| Commercial assay or kit | Pierce High Capacity Streptavidin Agarose | Pierce (Thermo Scientific) | #20359 | |
| Chemical compound, drug | PhosSTOP | Sigma-Aldrich | T10282 | |
| Chemical compound, drug | Puromycin | Gold Biotechnology | Gold Biotechnology | |
| Chemical compound, drug | DAPI | Thermo Fisher Scientific | D1306 | |
| Chemical compound, drug | Protease inhibitor cocktail | Sigma-Aldrich | P8340 | |
| Chemical compound, drug | TCEP | Gold Biotechnology | TCEP2 | |
| Chemical compound, drug | Formic Acid | Sigma-Aldrich | #94318 | |
| Peptide, recombinant protein | Trypsin | Promega | V511C | |
| Peptide, recombinant protein | Lys-C | | 129–02541 | |
| Commercial assay or kit | Trypan Blue Stain Thermo Fisher Scientific | Wako Chemicals | 129–02541 w | |
| Commercial assay or kit | BioRad Protein Assay Dye Reagent Concentrate | Bio-Rad | 5000006 | |
| Chemical compound, drug | Urea | Sigma | U5378 | |
| Chemical compound, drug | EPPS | Sigma-Aldrich | E9502 | |
| Chemical compound, drug | 2-Chloroacetamide | Sigma-Aldrich | C0267 | |
| Other | Empore SPE Disks C18 3 M | Sigma-Aldrich | #66883 U | |
| Commercial assay or kit | Pierce Quantitative Colorimetric Peptide Assay | Thermo Fisher Scientific | #23275 | |
| Recombinant DNA reagent | pHAGE-EGFP-NDP52 | *Heo et al., 2015* | Addgene #175749; RRID:Addgene_175749 | |
| Recombinant DNA reagent | pHAGE-EGFP-OPTN | *Heo et al., 2015* | Addgene #175750; RRID:Addgene_175750 | |
| Recombinant DNA reagent | pHAGE-APEX2-FLAG-GABARAPL2 | This paper | RRID:Addgene_175751 | Addgene #175751 |
| Recombinant DNA reagent | pHAGE-APEX2-FLAG-GABARAPL2$^{Y49A/L50A}$ | This paper | RRID:Addgene_175752 | Addgene #175752 |
| Recombinant DNA reagent | pHAGE-APEX2-FLAG-MAP1LC3B | This paper | RRID:Addgene_175753 | Addgene #175753 |

*Continued*

| Reagent type (species) or resource | Designation | Source or reference | Identifiers | Additional information |
|---|---|---|---|---|
| Recombinant DNA reagent | pHAGE-APEX2-FLAG-MAP1LC3B$^{K51A}$ | This paper | RRID:Addgene_175754 | Addgene #175754 |
| Recombinant DNA reagent | pHAGE-APEX2-FLAG-LGALS1 | This paper | RRID:Addgene_175755 | Addgene #175755 |
| Recombinant DNA reagent | pHAGE-APEX2-FLAG-LGALS3 | This paper | RRID:Addgene_175756 | Addgene #175756 |
| Recombinant DNA reagent | pHAGE-APEX2-FLAG-GFP | This paper | RRID:Addgene_175757 | Addgene #175757 |
| Recombinant DNA reagent | pHAGE-APEX2-FLAG-LGALS8 | This paper | RRID:Addgene_175758 | Addgene #175758 |
| Recombinant DNA reagent | pHAGE-APEX2-FLAG-CALCOCO2 | This paper | RRID:Addgene_175759 | Addgene #175759 |
| Recombinant DNA reagent | pHAGE-APEX2-FLAG-OPTN | This paper | RRID:Addgene_175760 | Addgene #175760 |
| Recombinant DNA reagent | pHAGE-APEX2-FLAG-TAX1BP1 | This paper | RRID:Addgene_175761 | Addgene #175761 |
| Recombinant DNA reagent | pHAGE-EGFP-OPTN D474N | This paper | RRID:Addgene_175762 | Addgene #175762 |
| Recombinant DNA reagent | pHAGE-EGFP-OPTN S473A 513 A | This paper | RRID:Addgene_175763 | Addgene #175763 |
| Recombinant DNA reagent | pHAGE-EGFP-OPTN S513A | This paper | RRID:Addgene_175764 | Addgene #175764 |

| Reagent type (species) or resource | Designation | Source or reference | Identifiers | Additional information |
|---|---|---|---|---|
| Recombinant DNA reagent | pHAGE-EGFP-OPTN E50K | This paper | RRID:Addgene_175765 | Addgene #175765 |
| Recombinant DNA reagent | pHAGE-EGFP-TAX1BP1 | This paper | RRID:Addgene_175766 | Addgene #175766 |
| Recombinant DNA reagent | pHAGE-EGFP-TAX1BP1 A114Q | This paper | RRID:Addgene_175767 | Addgene #175767 |
| Recombinant DNA reagent | pHAGE-EGFP-TAX1BP1 SKICH (1-140Δ) | This paper | RRID:Addgene_175768 | Addgene #175768 |
| Recombinant DNA reagent | pHAGE-EGFP-TAX1BP1 V192S | This paper | RRID:Addgene_175769 | Addgene #175769 |
| Recombinant DNA reagent | pHAGE-EGFP-TAX1BP1 Q770A E774K | This paper | RRID:Addgene_175770 | Addgene #175770 |
| Recombinant DNA reagent | pHAGE-EGFP-TAX1BP1 632-639Δ | This paper | RRID:Addgene_175771 | Addgene #175771 |
| Recombinant DNA reagent | pHAGE-EGFP-TAX1BP1 Y635A | This paper | RRID:Addgene_175772 | Addgene #175772 |
| Recombinant DNA reagent | pHAGE-EGFP-TAX1BP1 N637A | This paper | RRID:Addgene_175773 | Addgene #175773 |
| Recombinant DNA reagent | pHAGE-EGFP-TAX1BP1 CC1Δ | This paper | RRID:Addgene_175774 | Addgene #175774 |
| Recombinant DNA reagent | pHAGE-EGFP-TAX1BP1 CC2Δ | This paper | RRID:Addgene_175775 | Addgene #175775 |

*Continued*

| Reagent type (species) or resource | Designation | Source or reference | Identifiers | Additional information |
|---|---|---|---|---|
| Recombinant DNA reagent | pHAGE-EGFP-TAX1BP1 CC3Δ | This paper | RRID:Addgene_175776 | Addgene #175776 |
| Recombinant DNA reagent | pSMART Tmem192-3X HA (targeting vector for genomic tagging) | This paper | RRID:Addgene_175777 | Addgene #175777 |
| Recombinant DNA reagent | pAC150 RFP-EGFP-LGALS3 | This paper | RRID:Addgene_175778 | Addgene #175778 |
| Recombinant DNA reagent | pAC150 RFP-EGFP-LGALS3 R186S | This paper | RRID:Addgene_175779 | Addgene #175779 |
| Recombinant DNA reagent | pHAGE-mKeima-LGALS3 | This paper | RRID:Addgene_175780 | Addgene #175780 |
| Recombinant DNA reagent | pCMV-hyBase-hyperactive piggyBac | *Yusa et al., 2011* | | |
| Software, algorithm | Prism | GraphPad, V9 | https://www.graphpad.com/scientificsoftware/ prism/ | |
| Software, algorithm | SEQUEST | *Eng et al., 1994* | N/A | |
| Software, algorithm | Flowjo | Flowjo, v10.7 | https://www.flowjo.com | |
| Software, algorithm | Perseus | Perseus v1.6.15.0 *Tyanova et al., 2016* | https://maxquant.org/perseus/ | |
| Software, algorithm | Fiji | ImageJ V.2.0.0 | https://imagej.net/software/fiji/ | |
| Software, algorithm | Imagelab | BioRad, v6.0.1 | https://www.bio-rad.com/en-us/product/image-lab-software?ID=KRE6P5E8Z&source_wt=imagelabsoftware_surl | |
| Software, algorithm | Cell Profiler | CellProfiler v4.0.6 | https://cellprofiler.org/ | |
| Software, algorithm | Metamorph | Metamorph v | https://www.moleculardevices.com/products/cellular-imaging-systems/acquisition-and-analysis-software/metamorph-microscopy#gref | |

## Cell culture

All assays performed in *Figures 1–3* were performed in HeLa cells (ATCC). Keima flux assays in *Figures 4–6* and *Figure 8* were performed Hela Flip-In T-Rex (HFT) cells (Brian Raught, Ontario Cancer Institute) and have been previously described in *Heo et al., 2015* or HeLa cells as indicated. HeLa and HFT cells were grown in Dulbecco' modifies Eagles medium (DMEM) supplemented with 10 % fetal bovine serum (FBS) with 5 % penicillin–streptomycin. Stable cell lines were generated using lentivirus generated from HEK293T. Antibiotic selections were performed with 1 µg/ml puromycin, 10 µg/ml blasticidin, or 100 µg/ml hygromycin. Cells were subjected to karyotyping (GTG-banded karyotype) by Brigham and Women's Hospital Cytogenomics Core Laboratory. H9 ES cells were from WiCell, who provides original cell stocks. All cell lines were found to be free of mycoplasma using Mycoplasma Plus PCR assay kit (Agilent).

## Neural differentiation of AAVS1-TRE3G-NGN2 pluripotent stem cells

A detailed version of this protocol is available at dx.doi.org/10.17504/protocols.io.br9em93e. Briefly, human ES cells (H9, WiCell Institute) with TRE3G-NGN2 integrated into the AAVS site have been previously described (*Ordureau et al., 2020*) and were cultured in E8 medium on Matrigel coated plates. To generate induced neurons (i3-neurons) from ES cells, cells were plated at $2 \times 10^5$ cells/ml on Day 0 on plates coated with Matrigel in ND1 medium (DMEM/F12, 1× N2 (thermo), human brain-derived neurotrophic factor (BDNF) (10 ng/ml, PeproTech)), human Neurotrophin-3 NT3 (10 ng/ml, PeproTech), 1 × nonessential amino acids, Human Laminin (0.2 µg/ml) ,and doxycycline (2 µg/ml). The media was replaced with ND1 the next day. On the next day, the medium was replaced with ND2 neurobasal medium, 1 × B27, 1× Glutamax, BDNF (10 ng/ml), NT3 (10 ng/ml), and doxycycline at 2 µg/ml. On Days 4 and 6, 50 % of the media was changed with fresh ND2. On Day 7, cells were

replated at 4 × 10$^5$ cells/well in ND2 medium supplemented with Y27632 (rock inhibitor – 10 μM). The media was replaced the next day with fresh ND2 and on Day 10 onwards 50 % media change was performed until the experimental day (Day 14 of differentiation unless otherwise noted).

## Imaging

A detailed version of this protocol is available on protocols.io at dx.doi.org/10.17504/protocols.io.bxghpjt6. Briefly, cells were plated onto 35-mm glass bottom dish (No. 1.5, 14 -mm glass diameter, MatTek). Live cells were imaged at 37 °C in prewarmed Fluorobrite supplemented with 10 % FBS. For all immunofluorescence experiments, cells were first fixed at room temperature with 4 % paraformaldehyde plus in PBS, solubilized in 0.1 % Triton-X in PBS blocked with 1 % BSA/0.1 % Triton-X in PBS and then immunostained. Primary antibodies were used at 1:500 and AlexaFluor-conjugated antibodies (Thermo Fisher) were used at 1:300. Images of fixed cells were captured at room temperature. Cells were imaged using a Yokogawa CSU-X1 spinning disk confocal on a Nikon Ti-E inverted microscope at the Nikon Imaging Center in Harvard Medical School. Nikon Perfect Focus System was used to maintain cell focus over time. The microscope equipped with a Nikon Plan Apo 40 ×/1.30 N.A. or 100 x/1.40 N.A. objective lens and 445 nm (75 mW), 488 nm (100 mW), 561 nm (100 mW), and 642 nm (100 mW) laser lines controlled by AOTF. Pairs of images for ratiometric analysis of mKeima fluorescence were collected sequentially using 100 mW 442 nm and 100 mW 561 solid-state lasers and emission collected with a 620/60 nm filter (Chroma Technologies). All images were collected with a Hamamatsu ORCA-ER cooled CCD camera (6.45 μm$^2$ photodiode) with MetaMorph image acquisition software. Z series are displayed as maximum *z*-projections and brightness and contrast were adjusted for each image equally and then converted to rgb for publication using FiJi software. Image analysis was performed using both Fiji and Cell Profiler (*McQuin et al., 2018*). Mander's Overlap Correlation (MOC) in lysosomes was performed in Cell Profiler. Each field of view for every unique condition was thresholded in the same way with a consistent pipeline. The 'identify primary objects' tool was used to find puncta for both the lysosome channel and for the respective receptor or p-TBK1 stain. The 'measure colocalization' module was used to compare the fluorescence intensities within the areas defined by the threshold. The MOC with Costes was reported for each field of view.

The LGALS3 puncta detected by immunofluorescence in HeLa cells and the RFP-GFP-LGALS3 puncta detected in the iN system upon LLOMe treatment and subsequent washout were all identified using Cell Profiler and the following protocols available on protocols.io at dx.doi.org/10.17504/protocols.io.bx8pprvn (for HeLa cells LGALS3) and dx.doi.org/10.17504/protocols.io.bx48pqzw (for iN RFP-GFP-LGALS3). Each cell area was first defined using a using a 'identify primary objects' module that included objects 200–1000 pixels units, and each punctum was marked using a 'identify primary objects' module that included objects 2–20 pixels units both with an optimized 'robust background' threshold. Each cell for each condition was thresholded in the same way with a consistent pipeline. Object size and shape were measured, and each punctum was related to its respective cell to yield a punctum per cell readout.

## Analysis of lysophagic flux in cultured cells Using Lyso-Keima

A detailed protocol for analysis of lysophagic flux by western blotting, flow cytometry, and imaging using Lyso-Keima is available on protocols.io at dx.doi.org/10.17504/protocols.io.bx8qprvw.

### Lysophagy assays

Lysophagy assays were carried out as previously described using as described in *Maejima et al., 2013* with slight modifications. HFT cells or iNeurons were treated with DMEM or ND2 containing 500–1000 μM LLOMe (L-Leucyl-L-Leucine methyl ester hydrochloride, Cayman Chemical) or 200 μM GPN for 1 hr, then media was replaced with fresh DMEM (referred to as 'washout' in the text). The cells were collected at the indicated time points after the LLOMe washout for various downstream assays.

### Western blotting

For western blotting, cell pellets were collected and resuspended in 8 M urea buffer (8 M urea, 150 mM TRIS pH, NaCl) supplemented with Protease and Phosphatase Inhibitors. The resuspended pellets were sonicated and the lysate was spun at 13,000 rpm for 10 min. Bradford or BCA assay was

performed on clarified lysate and equal amounts of lysate were boiled in 1× SDS containing Laemmeli buffer. Lysates were run on 4%–20% Tris Glycine gels (BioRad) and transferred via Wet transfer onto PVDF membranes for immunoblotting with the indicated antibodies. Images of blots were acquired using Enhanced-Chemiluminescence on a BioRad ChemiDoc imager.

### Flow cytometry
Cells of the indicated genotypes were grown to 70 % confluency in 6-well plates and then treated with various drugs for the indicated time points. At the time of harvesting, cells were trypsinized, pelleted at 1000 rpm for 3 min, and then resuspended in FACS buffer (1× PBS, 2 % FBS). The resuspend cells were filtered through cell strainer caps into FACS tubes (Corning, 352235) and placed on ice. The cells (~10,000 per replicate) were then analyzed by flow cytometry on a BD FACSymphony flow cytometer and the data were exported into Flowjo. After gating for live, single cells and Keima-positive cells, the 561 nm (acidic) to 488 nm (neutral) excitation ratio was calculated in Flowjo by diving the mean values of 561 nm excited cells to those excited at 488 nm.

### Imaging
Analysis of acidic Keima-LGALS3 puncta at 12 hr washout was done in Cell Profiler using a consistent pipeline for each condition. The 'image math' module where the 561 nm-excitation channel image was divided by the 442 nm-excitation channel image. The acidic puncta in the resulting image were marked using the 'identify primary objects' tool by applying an Otsu threshold for puncta 5–20 pixels in diameter. Each resulting punctum was matched to its respective cell and counted. The 'image math' image was exported, and a 'Fire'look up table in Fiji was applied to show the acidic signal (561 nm/442 nm) hotspots. An image of the acidic puncta identified was also exported with each punctum having a separate color.

## Gene editing
Gene editing in HFT and HeLa cells was performed as described in *Ran et al., 2013*. Gene editing in H9 ES cells was performed as *Ordureau et al., 2020* in HFT cells lacking TBK1 or TKO (CALCOCO2$^{-/-}$, OPTN$^{-/-}$, TAX1BP1$^{-/-}$) were described in *Heo et al., 2015*. Guide sequences used were as follows: TBK1 (Exon 1, 5'-AGACATTTGCAGTAGCTCCT -3'); OPTN (Exon 1, 5'-AAACCTGGACAC-GTTTACCC-3'); NDP52 (Exon 1, 5'-GGATCACTGTCATTTCTCTC-3'); TAX1BP1 (Exon 2, 5'-CCACATC-CAAAAGATTGGGT-3'); SQSTM1 (Exon 2, 5'-CGACTTGTGTAGCGTCTGCG-3'); ATG7 (Exon 1, 5'-ATCCAAGGCACTACTAAAAG-3'); OPTN (Exon 1, 5'-AAACCTGGACACGTTTACCC-3'); CALCOCO2 (Exon 1, 5'-GGATCACTGTCATTTCTCTC-3'); ATG5 (Exon 5, 5' -GATCACAAGCAACTCTGGAT-3'); TMEM192 (Exon 4, 5'-AGTAGAACGTGAGAGGCTCA-3'). To engineer the Lyso-IP tag into HeLa cells using homology-directed repair, a gene block encoding a 3xHA epitope tag, a puromycin cassette, and homology arms on either side of the cleavage site was synthesized by Integrated DNA Technologies to edit the *tmem192* locus. This sequence was cloned into the pSmart (Lucigen Cat#40041–2) shuttle vector using Gibson assembly (New England Biolabs). The shuttle vector along with the TMEM192 sgRNA sequence was transiently transfected into HeLa cells and puromycin selection was performed 5 days post-transfection for 7–8 days. The mixed pool of cells that were puromycin resistant were single-cell plated and clonal lines of homozygous HeLa TMEM192$^{HA}$ were isolated.

Gene editing in ES cells was performed as in *Ordureau et al., 2018*. Guide RNAs were generated using the GeneArt Precision gRNA Synthesis Kit (Thermo Fisher Scientific) according to the manufacturer's instruction and purified using RNeasy Mini Kit (QIAGEN). 0.6 µg sgRNA was incubated with 3 µg SpCas9 protein for 10 mins at room temperature and electroporated into $2 \times 10^5$ H9 cells using Neon transfection system (Thermo Fisher Scientific). Out of frame deletions were verified by DNA sequencing with Illumina MiSeq and by immunoblotting.

## Molecular cloning
Stable expression plasmids were generally made using either Gateway technology (Thermo) or via Gibson assembly (New England biolabs) in pHAGE backbone unless otherwise noted. Entry clones from the human orfeome collection version eight were obtained and cloned via LR cloning into various destination expression vectors. Site-directed mutagenesis was carried out using the Quick-Change Site Directed Mutagenesis Kit (New England Biolabs) as per the manufacture's instructions.

## Stable cell line generation

Lentiviral vectors were packaged in HEK293T by cotransfection of pPAX2, pMD2, and the vector of interest in a 4:2:1 ratio using polyethelenimine. Virus containing supernatant was collected 2 days after transfection and filtered through 0.22- µm syringe filter. Polybrene was added at 8 µg/ml to the viral supernatant. After infecting target cells with varying amounts of relevant viruses, cells were selected in puromycin (1 µg/ml), blasticidin (10 µg/ml), or hygromycin (100 µg/ml). In case of GFP expressing lines, further selection was carried out using FACS for GFP-positive cells.

## Lysosomal immunoprecipitation

A protocol for analysis of Lyso-IP by proteomics is provided at protocols.io: dx.doi.org/10.17504/protocols.io.bw7hphj6 and is a modification of a related protocol (dx.doi.org/10.17504/protocols.io.bybjpskn). Lysosomal immunoprecipitation was carried out as described in *Wyant et al., 2018* with a few modifications. Briefly, cells endogenously tagged with TMEM192$^{HA}$ were seeded in 15- cm plates. All buffers were supplemented with protease inhibitors. At 80 % confluency the cells were harvested on ice by scraping and washed once with PBS containing protease inhibitors (Roche). The cells were pelleted at 300 g for 5 min at 4°C and were washed once with KPBS buffer (136 mM KCl, 10 mM KH$_2$PO$_4$, 50 mM sucrose, pH 7.2). The cell pellet was resuspended in 1 ml KPBS and lysed using 30 strokes in a 2- ml Potter-Elvehjem homogenizer. The lysed cells were spun down at 1000 g for 5 min at 4 °C. The pellet was discarded and the protein concentration of the lysate was determined by Bradford assay. After normalizing the protein concentration to be equal across all replicates, 5 % of the input sample was saved and 50–100 µl of anti-HA magnetic beads was added the remainder of the sample. This mixture was placed on gentle rotation for 20 min, and beads were separated from the lysate using a magnetic stand. The beads were washed twice with KPBS containing 300 mM NaCl and once with KPBS buffer. The samples were then eluted either by boiling the beads with 100 µl 2 × Laemmli buffer (for western blot) for 10 min or with 100 µl KPBS containing 0.5% NP-40 in thermo mixer at 30 °C for 20 min (for mass spectrometry). Elutes for mass spectrometry were snap frozen in liquid nitrogen and stored in –80 °C until further processing.

## Quantitative proteomics

A detailed version of this protocol is available at dx.doi.org/10.17504/protocols.io.bw7hphj6.

Sample preparation for mass spectrometry-lysosomal fractions. For mass spectrometry of lyso-somal eluates, samples were reduced using TCEP (5 mM for 10 min at 55°C) and alkylated (with chloroacetamide 20 mM at room temperature for 30 min) prior to TCA precipitation. TCA was added to eluates at final concentration of 20 % and placed on ice at 4°C for at least an hour. Precipitates were pelleted for 30 min at maximum speed at 4°C, and then the pellets were washed three times using ice cold methanol. Dried pellets were then resuspended in in 50 µl, 200 mM EPPS, pH 8.0. Peptide digestion was carried out using LysC (Wako cat. # 129-02541, 0.25 µg) for 2 h at 37°C followed by trypsin (0.5 µg) overnight. Digested peptides were then labelled with 4 µl of TMT reagent (at 20 µg/µl stock) for 1 hr and the reaction was quenched using hydroxylamine at a final concentration of 0.5 % (wt/vol) for 20 min. The samples were the combined and dried in a vacuum centrifuge. This combined sample was then subjected to fractionation using the high pH reversed-phase peptide fractionation kit (Thermo Fisher) for a final of six fractions. The dried fractions were processed by C18 stage tip desalting prior mass spectrometry.

Sample preparation for Mass Spectrometry-APEX2 Proteomics. For APEX2 proteomics, cells expressing various APEX2 fusions were processed as in *Heo et al., 2018*; *Hung et al., 2016*. To induce proximity labeling in live cells, cells were incubated with 500 µM biotin phenol (LS-3500.0250, Iris Biotech) for 1 hr and treated with 1 mM H$_2$O$_2$ for 1 min, and the reaction was quenched with 1× PBS supplemented with 5 mM Trolox, 10 mM sodium ascorbate, and 10 mM sodium azide. Cells were then harvested, and lysed in radioimmunoprecipitation assay (RIPA) buffer (supplemented with 5 mM Trolox, 10 mM sodium ascorbate, and 10 mM sodium azide). To enrich biotinylated proteins, an identical amount of cleared lysates in each cell was subjected to affinity purification by incubating with the streptavidin-coated magnetic beads (catalog no. 88817, Pierce) for 1 hr at room temperature. Beads were subsequently washed twice with RIPA buffer, once with 1 M KCl, once with 0.1 M NaCO$_3$, once with 2 M urea, twice with RIPA buffer, and three times with PBS.

For proteomics, biotinylated protein bound to the beads was digested with trypsin in 0.1 M EPPS [4-(2-hydroxyethyl)-1-piperazinepropanesulfonic acid, 4-(2-hydroxyethyl)piperazine-1-propanesulfonic acid, *N*-(2-hydroxyethyl)piperazine-*N'*-(3-propanesulfonic acid)] (pH 8.5) overnight at 37 °C. To quantify the relative abundance of individual protein across different samples, each digest was labeled with TMT reagents (Thermo Fisher Scientific), mixed, and desalted with a C18 StageTip (packed with Empore C18; 3 M Corporation) before SPS-MS$^3$ analysis on an Orbitrap Lumos (Thermo Fisher Scientific) coupled to a Proxeon EASY-nLC1200 liquid chromatography (LC) pump (Thermo Fisher Scientific). Peptides were separated on a 100- μm inner diameter microcapillary column packed in house with ~35 cm of Accucore150 resin (2.6 μm, 150 Å, Thermo Fisher Scientific, San Jose, CA) with a gradient consisting of 5%–21% (ACN, 0.1 % FA) over a total 150 min run at ~500 nl/min (*McAlister et al., 2014*). Details of instrument parameters for each experiment are provided below.

For Multi-Notch MS$^3$-based TMT analysis (*McAlister et al., 2014*; *Paulo et al., 2016*), the scan sequence began with an MS$^1$ spectrum (Orbitrap analysis; resolution 60,000 at 200 Th; mass range 375–1500 m/z; automatic gain control (AGC) target $5 \times 10^5$; maximum injection time 50 ms) unless otherwise stated in the instrument parameters in each supplemental table (*Supplementary files 1–5*). Precursors for MS$^2$ analysis were selected using a Top10 method. MS$^2$ analysis consisted of collision-induced dissociation (quadrupole ion trap analysis; Turbo scan rate; AGC $2.0 \times 10^4$; isolation window 0.7 Th; normalized collision energy [NCE] 35; maximum injection time 90 ms). Monoisotopic peak assignment was used and previously interrogated precursors were excluded using a dynamic window (150 s ± 7 ppm) and dependent scans performed on a single charge state per precursor. Following acquisition of each MS$^2$ spectrum, a synchronous-precursor-selection (SPS) MS$^3$ scan was collected on the top 10 most intense ions in the MS$^2$ spectrum (*McAlister et al., 2014*). MS$^3$ precursors were fragmented by high-energy collision-induced dissociation and analyzed using the Orbitrap (NCE 65; AGC $3 \times 10^5$; maximum injection time 150 ms, resolution was 50,000 at 200 Th).

For proteomics data analysis, raw mass spectra obtained were processed as described in *Huttlin et al., 2010*; *Paulo et al., 2015*; *Ordureau et al., 2020* and were processed using a Sequest. Mass spectra were converted to mzXML using a version of ReAdW.exe. Database searching included all entries from the Human Reference Proteome. Searches were performed with the following settings (1) 20 ppm precursor ion tolerance for total protein level analysis, (2) product ion tolerance was set at 0.9 Da, (3) TMT or TMTpro on lysine residues or N-termini at +229.163 Da or +304.207 Da, and (4) carbamidomethylation of cysteine residues ( + 57.021 Da) as a static modification and oxidation of methionine residues ( +15.995 Da) as a variable modification. Peptide-spectrum matches (PSMs) were adjusted to a 1 % false discovery rate (*Elias and Gygi, 2007*). PSM filtering was performed using a linear discriminant analysis, as described previously (*Huttlin et al., 2010*). To quantify the TMT-based reporter ions in the datasets, the summed signal-to-noise (S:N) ratio for each TMT channel was obtained and found the closest matching centroid to the expected mass of the TMT reporter ion (integration tolerance of 0.003 Da). Proteins were quantified by summing reporter ion counts across all matching PSMs, as described previously (*Huttlin et al., 2010*). PSMs with poor quality, or isolation specificity less than 0.7, or with TMT reporter summed signal-to-noise ratio that were less than 100 or had no MS$^3$ spectra were excluded from quantification. Classification of proteins to various organellar locations or functional groups were performed using manually curated databases from Uniprot and are listed in the relevant supplementary tables (*Supplementary files 1–6*).

Values for protein quantification were exported and processed using Perseus to calculate Log FCs and p-values. Volcano plots using these values were plotted in Excel. The mass spectrometry proteomics data have been deposited to the ProteomeXchange Consortium via the PRIDEpartner repository (*Perez-Riverol et al., 2019*) with the dataset identifier PXDO27476.

## Statistics
All statistical data were calculated using GraphPad Prism v7 or Perseus. Comparisons of data were performed by one-way ANOVA with Tukey's multiple comparisons test.

## Acknowledgements
This work was supported by Aligning Science Across Parkinson's (ASAP) (JWH), the NIH (R01 NS083524, R01 NS110395 to JWH and K01DK098285 to JAP), the Harvard Medical School Cell Biology Initiative for Molecular Trafficking and Neurodegeneration, and a generous gift from Ned

Goodnow (JWH). Michael J Fox Foundation administers the grant ASAP-000282 on behalf of ASAP and itself. For the purpose of open access, the author has applied a CC-BY public copyright license to the Author Accepted Manuscript (AAM) version arising from this submission. VE and MJH were supported by Jane Coffin Childs fellowships. SS was supported by a Canadian Institutes of Health Research fellowship. We thank the Nikon Imaging Center at Harvard Medical School for microscopy support and Julia Paoli for technical support.

## Additional information

### Competing interests

J Wade Harper: Reviewing editor, eLife. Co-Founder and Scientitic Advisor for Caraway Therapeutics. Founding Advisory Board member for Interline Therapeutics. The other authors declare that no competing interests exist.

### Funding

| Funder | Grant reference number | Author |
|---|---|---|
| Aligning Science Across Parkinsons (ASAP) | ASAP-000282 | J Wade Harper |
| National Institutes of Health | NS083524 | J Wade Harper |
| National Institutes of Health | NS110395 | J Wade Harper |
| National Institutes of Health | DK098285 | Joao A Paulo |
| Jane Coffin Childs Memorial Fund for Medical Research | | Melissa J Hoyer Vinay V Eapen |
| Canadian Institutes of Health Research | | Sharan Swarup |
| Ned Goodnow Fund | | J Wade Harper |

The funders had no role in study design, data collection and interpretation, or the decision to submit the work for publication.

### Author contributions

Vinay V Eapen, Conceptualization, Data curation, Formal analysis, Funding acquisition, Investigation, Methodology, Validation, Visualization, Writing – original draft, Writing – review and editing; Sharan Swarup, Conceptualization, Data curation, Formal analysis, Funding acquisition, Investigation, Methodology, Validation, Visualization, Writing – review and editing; Melissa J Hoyer, Conceptualization, Formal analysis, Funding acquisition, Investigation, Methodology, Validation, Visualization, Writing – review and editing; Joao A Paulo, Data curation, Funding acquisition, Investigation, Methodology; J Wade Harper, Conceptualization, Formal analysis, Funding acquisition, Investigation, Project administration, Supervision, Visualization, Writing – original draft, Writing – review and editing

### Author ORCIDs

Vinay V Eapen ![ORCID] http://orcid.org/0000-0002-8023-387X
Sharan Swarup ![ORCID] http://orcid.org/0000-0003-0226-5582
Melissa J Hoyer ![ORCID] http://orcid.org/0000-0003-0858-4998
J Wade Harper ![ORCID] http://orcid.org/0000-0002-6944-7236

### Decision letter and Author response

Decision letter https://doi.org/10.7554/eLife.72328.sa1
Author response https://doi.org/10.7554/eLife.72328.sa2

## Additional files

### Supplementary files

• Supplementary file 1. Quantitative proteomic analysis of lysosomes purified from HeLa cells. See *Supplementary file 6* for details.

• Supplementary file 2. Quantitative proteomic analysis of lysosomes in response to lysosomal damage with GPN (time course). See *Supplementary file 6* for details.

• Supplementary file 3. Proximity biotinylation of APEX1-GABARAPL2 and APEX2-MAP1LC3B in response to lysosomal damage with LLOMe. See *Supplementary file 6* for details.

• Supplementary file 4. Proximity biotinylation of APEX1-LGALS1, LGALS3, LGALS8 and CALCOCO2 in response to lysosomal damage with LLOMe. See *Supplementary file 6* for details.

• Supplementary file 5. Proximity biotinylation of APEX1-OPTN, TAX1BP1, and SQSTM1 in response to lysosomal damage with LLOMe. See *Supplementary file 6* for details.

• Supplementary file 6. List of corresponding experiments for .RAW files submitted to ProteomeXchange Consortium via the PRIDEpartner respository.

• Transparent reporting form

### Data availability

All proteomic RAW files have been deposited in the PRIDE component of Proteome xchange with the identifier PXDO27476, and will be released upon publication.

The following dataset was generated:

| Author(s) | Year | Dataset title | Dataset URL | Database and Identifier |
|---|---|---|---|---|
| Joao P, Wade H | 2021 | Quantitative proteomics reveals the selectivity of ubiquitin-binding autophagy receptors in the turnover of damaged lysosomes by lysophagy | https://www.ebi.ac.uk/pride/archive/projects/PXD027476 | PRIDE, PXD027476 |

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
