## [Decision Letter]

**Acceptance summary:**

The manuscript identifies the molecular mechanisms that are instrumental in recovery of lysosomes after lysosome damage. It is a very rich resource of proteomic data and it provides evidence for the importance of the autophagy receptor TAX1BP1 and the protein kinase TBK1 for lysophagy flux. The authors demonstrate that lysosomes recover from damage by autophagosomal fusion of damaged organelles with intact healthy lysosomes.

**Decision letter after peer review:**

Thank you for submitting your article "Quantitative proteomics reveals the selectivity of ubiquitin-binding autophagy receptors in the turnover of damaged lysosomes by lysophagy" for consideration by *eLife*. Your article has been reviewed by 2 peer reviewers, including Ivan Dikic as Reviewing Editor and Reviewer #1, and the evaluation has been overseen by Suzanne Pfeffer as the Senior Editor. The following individual involved in review of your submission has agreed to reveal their identity: Geoffrey Hesketh (Reviewer #2).

This paper will be of significant interest to cell biologists studying lysophagy and selective autophagy more generally. It presents an extensive proteomic dataset, provides novel biological insight, and offers new reagents that will be of great value to the community. The data are of high quality and the conclusions are well supported.

Comments for authors to consider:

1) It is interesting that TBK1 seems to be essential for maintaining lysophagy flux in lyso-Keima assays. Did the authors find TBK1 in any of proximity biotinylation studies of the damaged lysosome? It seems to be minorly (<0.5 log2fold change) enriched in APEX-LC3B interactome in Figure 2C.

2) Is the Flow cytometry Keima-LGALS3 assay in Figure 8C similar when TAX1BP KO cells are used instead of triple KOs? This would indicate that lysophagy flux is strongly dependent on the SKICH domain of TAX1BP1 with very little contribution from CALCOCO1 and OPTN1. Does deletion of SKICH domain affect lysosomal localisation of TAX1BP1?

3) Authors suggest that lysosomes recover by fusion with healthy lysosomes. Is this due to an increase in lysosomal biogenesis which is triggered after Llome treatment? Since all the flux measurements are taken at 10-12h after Llome removal, lysosomal biogenesis (through TFEB) may be important in this case. The authors can discuss such possibilities in the Discussion section.

4) The authors adapted a LysoIP system in which endogenous TMEM192 is fused to a 3xHA tag, allowing affinity purification and subsequent mass spectrometric analysis of lysosomes under different cellular conditions. They also explored the lysosomal proteome by performing proximity-dependent biotinylation and mass spectrometry experiments using APEX2 – a peroxidase-based enzyme – fused to proteins that are recruited to damaged lysosomes. Using these proteomic tools in the context of lysosomal damage – induced by treatment with either GPN or LLoMe – the authors comprehensively explored the differential recruitment of proteins to damaged lysosomes, producing what they refer to as the "lysophagy proteome landscape". While of undeniable quality and value as a dataset, the authors did not specifically highlight novel proteins or potential mechanisms that their dataset revealed. Instead, subsequent functional analysis was focused on proteins with known roles in selective autophagy and / or lysophagy. In fact, the autophagy receptors that were investigated functionally were not particularly highly enriched in their "lysophagy proteome landscape" relative to many other proteins. Although not a major weakness, some specific discussion of potentially novel observations, if any, within the proteomic dataset would have been welcome. Nonetheless, despite not being a clear extension of the proteomic data, the subsequent functional analysis is of very high quality.

*Reviewer #1:*

In this manuscript Eapen et al., uses proximity biotinylation proteomics of autophagy adaptors, cargo receptors, and Galectins to monitor events following acute lysosomal damage induced by LLome and GPN treatment. The manuscript is a rich resource of proteomic data and identifies the different molecular players that are instrumental in recovery of lysosomes after Llome mediated damage. The authors nicely show the importance of TAX1BP1 and TBK1 in measuring lysophagy flux using the lyso-Keima reporter system and suggest that lysosomes recover from damage by autophagosomal fusion of damaged organelles with intact healthy lysosomes.

*Reviewer #2:*

In this paper, Eapen et al., investigate the molecular underpinnings of lysophagy – the clearance of damaged lysosomes through selective autophagy. The paper is split into two major parts. First, the authors perform a comprehensive series of proteomics experiments to investigate the differential recruitment of proteins to the lysosome upon damage. Next the authors undertake a detailed investigation of the functional requirement for different autophagy receptors in the process of lysophagy. Functional studies are carried out in HeLa cells, as well as preliminary investigations in induced Neurons (iNeurons). In addition to the creation of a valuable proteomic dataset that will fuel future studies into lysophagy, the authors demonstrate a specific function for the autophagy receptor TAXBP1 in lysophagy, and further define some of the molecular features of TAXBP1 that underlie its role in this process. Broadly speaking this study is extremely comprehensive, the data are of high quality, and the conclusions are well supported by the data. The proteomic dataset, the newly developed reagents and assays, and the mechanistic insights uncovered in the paper make important contributions to the selective autophagy field.

Key achievements in the paper are further summarized below.

1) The authors adapted a system previously developed in the Sabatini lab (which they call Lyso-IP) in which endogenous TMEM192 is fused to a 3xHA tag, allowing affinity purification and subsequent mass spectrometric analysis of lysosomes under different cellular conditions. They also explored the lysosomal proteome by performing proximity-dependent biotinylation and mass spectrometry experiments using APEX2 – a peroxidase-based enzyme – fused to proteins that are recruited to damaged lysosomes. Using these proteomic tools in the context of lysosomal damage – induced by treatment with either GPN or LLoMe – the authors comprehensively explored the differential recruitment of proteins to damaged lysosomes, producing what they refer to as the "lysophagy proteome landscape". While of undeniable quality and value as a dataset, the authors did not specifically highlight novel proteins or potential mechanisms that their dataset revealed. Instead, subsequent functional analysis was focused on proteins with known roles in selective autophagy and / or lysophagy. In fact, the autophagy receptors that were investigated functionally were not particularly highly enriched in their "lysophagy proteome landscape" relative to many other proteins. Although not a major weakness, some specific discussion of potentially novel observations, if any, within the proteomic dataset would have been welcomed. Nonetheless, despite not being a clear extension of the proteomic data, the subsequent functional analysis is of very high quality.

2) The authors systematically investigated the contributions of autophagy receptors and their regulation to the process of lysophagy induced by lysosomal damaging agents. They first developed a very nice reagent they call Lyso-Keima – a fusion between the galectin LGALS3 and the fluorophore mKeima, which when recruited to damaged lysosomes by LGALS3 undergoes an acidic pH-dependent spectral shift – to monitor lysophagic flux using different readouts: microscopy, flow cytometry, or Western blotting. Using the Lyso-Keima reagent in HeLa cell lines in which a variety of genetic perturbations and / or reconstitutions have been made, the authors demonstrated that the autophagy receptor TAXBP1 is both necessary and sufficient in lysophagy, with additional functional contribution from OPTN – though OPTN appears to be partially sufficient but not necessary. They also demonstrated a functional requirement for regulation through the kinase TBK1, although the critical substrates in lysophagy were not identified. More detailed analysis revealed a specific requirement for the TAXBP1 N-terminal SKICH domain in regulation of lysophagy, as well as a requirement for upstream ubiquitylation signals to induce TAXBP1 recruitment to damaged lysosomes.

3) Another nice feature of this paper is that the authors extended their observations made in HeLa cells to an induced neuron model system – iNeurons. Here they again showed a requirement for TBK1 and TAXBP1 in lysophagy. Their development of a lysophagy assay in iNeurons will be highly valuable for future studies investigating a potential role for lysophagy in cell-based models of neurodegenerative diseases.

---

## [Author Response]

Comments for authors to consider:1) It is interesting that TBK1 seems to be essential for maintaining lysophagy flux in lyso-Keima assays. Did the authors find TBK1 in any of proximity biotinylation studies of the damaged lysosome? It seems to be minorly (<0.5 log2fold change) enriched in APEX-LC3B interactome in Figure 2C.

This is an interesting point- We think that rather than enrichment of total TBK1 at the lysosome, it is actually a small pool of phosphorylated TBK1 (pTBK1) that is specifically being enriched during lysophagy. Given that multiple TBK1-associated cargo adaptors are undergoing basal autophagic flux, there is already a pool of TBK1 associated with the lysosome. Since we are not detecting peptides specific to pTBK1 in the proteomics, we wouldn’t necessarily expect a clear increase in total TBK1 in the proteomics, given how small the pool of pTBK1 is. We have demonstrated for mitophagy that less than 1% of TBK1 is phosphorylated at ~1h of depolarization, so we think it would be hard to visualize this by proteomics examining the total protein. This is why having the pTBK1 immunofluorescence data in the paper is so important. We hope that addresses the reviewer’s comment.

2) Is the Flow cytometry Keima-LGALS3 assay in Figure 8C similar when TAX1BP KO cells are used instead of triple KOs? This would indicate that lysophagy flux is strongly dependent on the SKICH domain of TAX1BP1 with very little contribution from CALCOCO1 and OPTN1. Does deletion of SKICH domain affect lysosomal localisation of TAX1BP1?

The reviewer brings up a good question. We performed the rescue experiment using the TKO background given the previous observation that physical interactions between cargo receptors (OPTN, CALCOCO2, and TAX1BP1) can occur under various conditions (both basally and under stimulation). As such, we felt that looking at these TAX1BP1 mutants without contributions of other interacting receptors would be the best way to look at the functions of the TAX1BP1 SKICH domain in isolation, for example.

In terms of the role of the SKICH domain in recruitment, we looked at the localization of the δ SKICH TAX1BP1 mutant and A114Q mutant and found that they both form punctate structures near or are in close proximity to damaged lysosomes. We have added the GFP-deltaSKICH and A114Q localization experiment to supplemental Figure S8. These results indicate that translocation per se of these mutants is not sufficient to explain their defect in lysophagic flux. Rather, the inability of this mutant to stimulate TBK1 activity and/or recruit the RB1CC1 complex at the lysosome likely explains the core defect.

3) Authors suggest that lysosomes recover by fusion with healthy lysosomes. Is this due to an increase in lysosomal biogenesis which is triggered after Llome treatment? Since all the flux measurements are taken at 10-12h after Llome removal, lysosomal biogenesis (through TFEB) may be important in this case. The authors can discuss such possibilities in the Discussion section.

Yes, this has been shown in the recent paper from the Ballabio Lab, which is now cited. We have now covered this point in the discussion and main text.

4) The authors adapted a LysoIP system in which endogenous TMEM192 is fused to a 3xHA tag, allowing affinity purification and subsequent mass spectrometric analysis of lysosomes under different cellular conditions. They also explored the lysosomal proteome by performing proximity-dependent biotinylation and mass spectrometry experiments using APEX2 – a peroxidase-based enzyme – fused to proteins that are recruited to damaged lysosomes. Using these proteomic tools in the context of lysosomal damage – induced by treatment with either GPN or LLoMe – the authors comprehensively explored the differential recruitment of proteins to damaged lysosomes, producing what they refer to as the "lysophagy proteome landscape". While of undeniable quality and value as a dataset, the authors did not specifically highlight novel proteins or potential mechanisms that their dataset revealed. Instead, subsequent functional analysis was focused on proteins with known roles in selective autophagy and / or lysophagy. In fact, the autophagy receptors that were investigated functionally were not particularly highly enriched in their "lysophagy proteome landscape" relative to many other proteins. Although not a major weakness, some specific discussion of potentially novel observations, if any, within the proteomic dataset would have been welcome. Nonetheless, despite not being a clear extension of the proteomic data, the subsequent functional analysis is of very high quality.

We thank the reviewer for highlighting our functional analysis and for the positive comments. We fully accept the point that we didn’t specifically follow up on the candidates that were generated in our lysophagy proteome landscape. We are hoping to prioritize these candidates with a subsequent CRISPR screen using our Lyso-Keima reagent. While some discussion of the potential novel candidates identified would be interesting, we feel that it would be premature to hypothesize their mechanism of action in the absence of any supporting data in the literature. However, we do agree that proteins with known roles in lysosomal/autophagy biology could be highlighted and discussed-we have tried to update our discussion to reflect these points. As to the point regarding the enrichment of known receptors in the datasets- it is well taken. We routinely do see some (CALCOCO2/SQSTM1) being significantly enriched in almost every proteome dataset, while others (TAX1BP1/OPTN) are usually enriched to a lower extent. This could be due to several reasons. (1) The stochastic nature of mass-spectrometry and also because peptides corresponding to some of these proteins (such as OPTN) may not fly well in the Mass-Spec. (2) Many selective autophagy receptors are being basally delivered to the lysosome for degradation therefore any “enrichment” at the lysosome might not be significant using our measurements. (3) The enrichment of specific receptors via mass spec in various forms of selective autophagy (for eg: mitophagy) does not always track with their relative importance in the pathway – SQSTM1 is always the most enriched protein in Mito preps but has no functional role in mitophagy. We think that more abundant Ub binding receptors may mask the effects of other similar proteins, but the results also emphasize the discordance between recruitment and functional importance, a point we now make in the discussion.